Follow Up

# Catalytic activity of KMT5B promotes ciliogenesis without affecting global chromatin accessibility

Janet Tait[1], Carmen Marthen[1], Barbara Hoelscher[1], Tobias Straub[1] , Ohnmar Hsam[1,2], Ralph AW Rupp[1]

**Multiciliated cells (MCCs) are specialized cells found in the brain, reproductive and respiratory tracts of mammals, and the epidermis of tadpole-stage *Xenopus* embryos. KMT5B and KMT5C are histone methyltransferases that deposit the dimethyl mark on histone 4 lysine 20 (H4K20). We previously showed that KMT5B/C double knockdown down-regulates H4K20me2 levels in bulk chromatin, as well as transcription of ciliary genes. MCCs of embryos lacking both enzymes, or only KMT5B, have depleted cilia. Here, we separate the function of KMT5B in multicilio-genesis and show that single knockdown of KMT5B, not KMT5C, leads to aberrant transcription and down-regulation of ciliary genes. This phenotype is rescued by catalytically active PHF8, an H4K20me1 demethylase, whereas hormone-inducible multicilin (MCI), master regulator of cilia, has no effect. Notably, the expression of key transcription factors of ciliogenesis is unaffected by KMT5B depletion, which dominates the transcriptional response to ectopic multicilin. Finally, ATAC-seq in animal caps shows KMT5B knockdown results in few differentially accessible peaks and does not compact chromatin at ciliary genes. This suggests KMT5B regulates MCCs via an alternative pathway to the canonical MCI-driven programme.**

## Introduction

Cellular differentiation is a dynamic process by which cells progress from pluripotency into their specific cell fate. Sometimes they acquire specialized structures and employ complex gene regulatory networks in the process of differentiation. Multiciliated cells (MCCs) are a highly specialized and post-mitotic cell type found in the mammalian brain, fallopian tubes, and bronchi (Bustamante-Marin & Ostrowski, 2017). These cells contain hundreds of motile cilia that beat in a coordinated manner to generate fluid flow. MCCs resembling those of the mammalian lungs are also found on the surface of the Xenopus embryonic epidermis (Walentek, 2021).

MCCs are specified by NOTCH signalling, which selects for MCC identity by inducing the expression of the geminin proteins multicilin (MCI), the so-called "master regulator" of ciliogenesis, and GEMC1 in humans (Stubbs et al, 2012; Ma et al, 2014; Meunier & Azimzadeh, 2016). From there, a cascade of transcription factors tightly controls ciliogenesis. Multicilin binds E2F4/5 transcription factors, which induces cell cycle exit in favour of differentiation. C-MYB, an S-phase protein, and CCNO, a cyclin-like protein, co-ordinate the formation of basal bodies, the specialized centrioles that nucleate cilia (Stubbs et al, 2012; Ma et al, 2014). Simultaneously, RFX family transcription factors alongside FOXJ1 control ciliary formation and the formation of the actin cap, a lattice of actin meshwork at the apical surface of MCCs (Quigley & Kintner, 2017). Our laboratory previously demonstrated an unexpected link between histone 4 lysine 20 (H4K20) methylating enzymes and MCCs in the Xenopus embryonic epidermis (Angerilli et al, 2023).

Histone methyltransferases are responsible for writing methyl marks on specific amino acid residues on histone tails. H4K20 can be unmethylated, or mono-, di-, or trimethylated. The marks are present at different abundance on the chromatin, with ~10% unmethylated, 10% H4K20me1, 80% H4K20me2—the highest abundance of any histone modification on the chromatin—and less than 1% H4K20me3 (Yang et al, 2008; Young et al, 2009; Corvalan & Coller, 2020; Pokrovsky et al, 2021). H4K20 marks are generated in a cell cycle–dependent manner, and there are a number of enzymes that target this site. H4K20me1 is written by SET8/PR-SET7 in the G2 phase and M phase, before proteolytic degradation of SET8 in the late G1 phase (Oda et al, 2009). H4K20me1 is then largely converted into H4K20me2 by KMT5B/SUV4-20H1 and KMT5C/SUV4-20H2 and further on to H4K20me3 by KMT5C (Schotta et al, 2008; Eid et al, 2016). H4K20 demethylases have also been identified. PHF8 converts H4K20me1 into unmethylated H4K20 (Liu et al, 2010; Qi et al, 2010), whereas RSBN1 has been shown to demethylate H4K20me2 (Brejc et al, 2017), and recently, hHR23b has been shown to have activity towards all H4K20me states in vitro (Cao et al, 2020).

H4K20 methylating and demethylating enzymes also play critical roles in controlling gene expression during embryonic development. SET8 depletion is embryonic lethal in mice and *Drosophila*

[1]Department of Molecular Biology, Biomedical Center, Ludwig-Maximilians-Universität München, Planegg-Martinsried, Germany   [2]Department of Neurology, University of Regensburg, Regensburg, Germany

Correspondence: ralph.rupp@bmc.med.lmu.de

 

(Schotta et al, 2008; Oda et al, 2009). Despite high sequence homology between KMT5B and KMT5C, they have diverse functions and roles in development. KMT5B is ubiquitously expressed during mouse development, whereas KMT5C abundance is lower and more tissue-specific. KMT5B knockout also results in perinatal lethality, whereas KMT5C knockout mice develop normally (Schotta et al, 2008). In *Xenopus*, double knockdown of KMT5B and KMT5C results in ectodermal defects including craniofacial abnormalities, loss of melanocytes, and reduced eye structures (Nicetto et al, 2013; Angerilli et al, 2023). KMT5B is also a well-known autism spectrum disorder risk factor gene (Wickramasekara & Stessman, 2019). PHF8 knockdown also has developmental consequences. In zebrafish, PHF8 knockdown leads to craniofacial defects and neurodevelopmental defects in the neural tube, and PHF8 is linked to Siderius X-linked intellectual disability (Qi et al, 2010).

We previously showed that double knockdown of KMT5B/C leads to a massive shift from H4K20me2 to H4K20me1 in bulk embryonic chromatin, coinciding with strong phenotypic and transcriptional effects on *Xenopus* MCCs, and furthermore, that knockdown of KMT5B alone can recapitulate the phenotypic changes in MCCs (Angerilli et al, 2023). Knockdown MCCs have much fewer, shorter cilia, a reduced apical actin cap, and, in some cases, clumped basal bodies. Here, we investigate this ciliogenic phenotype further and show that KMT5B knockdown alone has strong effects on gene expression, particularly expression of ciliogenic genes, in contrast to KMT5C, and that KMT5B knockdown drives transcriptional changes even in the presence of exogenous multicilin, using multicilin-overexpressing animal caps (ectodermal organoids). In addition, we find that rescue of cilia by PHF8 relies on the catalytic activity of the enzyme. Lastly, we show that KMT5B knockdown has no substantial effect on chromatin accessibility, either on ciliogenic genes or in general. Taken together, our data shed further light on the role of KMT5B in MCCs.

# Results

### Knockdown of KMT5B results in transcriptional changes

We performed confocal microscopy on embryos in which either KMT5B or KMT5C was knocked down by translation-blocking morpholinos (Fig 1A). Confirming previous whole-mount immunocytochemistry results, we see that knockdown of KMT5C has no effect on cilia, actin cap, or cell size (Fig S1). Knockdown of KMT5B results in fewer, shorter cilia, in a depleted actin cap, and occasionally in clumped basal bodies, close to the nucleus of MCCs (Fig S2), as had been described for KMT5B/C double knockdown (Angerilli et al, 2023). To determine the individual effects of depleting KMT5B and KMT5C on gene expression, we compared the gene expression profiles of KMT5B and KMT5C knockdown animal cap organoids at the neurula stage (NF16). Animal caps are explants taken from the blastocoel roof of blastula-stage embryos, and they differentiate by default into a mucociliary epithelium. Using animal caps instead of whole embryos results in a simplified tissue in which only five cell types are present instead of the

hundreds of cell types of a whole embryo. Our first observation was that very few genes were misregulated by KMT5C knockdown. Although more than 2,000 genes are misregulated in KMT5B knockdown animal caps, only 66 genes were misregulated in KMT5C knockdown animal caps compared with control morpholino-injected animal caps (Fig 1B, C, E). We attempted to perform GO analysis on the misregulated genes upon KMT5C knockdown and found no enriched GO categories for the up-regulated genes and only three enriched GO terms in the down-regulated genes: "regulation of transcription by RNA polymerase II," "circadian rhythm," and "circadian regulation of gene expression," each category containing very few genes. This is likely due to the overall small cohort of misregulated genes. None of these genes are involved in cell size regulation or actin remodelling. That KMT5C has such a weak effect on transcription is surprising because we previously showed that both KMT5B and KMT5C are responsible for writing H4K20me2 in the *Xenopus* embryo, and KMT5C further writes H4K20me3, a key heterochromatic mark.

KMT5B misregulation, on the other hand, led to 1,165 up-regulated genes and 1,505 down-regulated genes (Fig 1B and C). Immediately, a ciliogenic connotation stuck out in the misregulated genes. For example, among the top misregulated genes were cntln, a centriolar-linked protein that binds microtubules (Jing et al, 2016), intraflagellar transport protein IFT88, and DNAL4, a dynein protein found in motile cilia (Fig 1D). To our surprise, we found a tubulin protein, tubb4a, to be one of the most up-regulated genes upon KMT5B knockdown (Fig 1D). However, TUBB4A is one of two ß-tubulins that contain a C-terminal motif that allows them to interact with cilia, and although knockout of tubb4b leads to severe ciliogenic defects in the MCCS of the mouse airway, TUBB4A depletion does not affect ciliary tuft formation. We found tubb4b among the down-regulated genes upon KMT5B depletion. TUBB4A may also play a role in promoting proliferation (Zhang et al, 2023).

We went on to perform GO analysis, and confirmed a strong ciliogenic connotation in the down-regulated genes. The top misregulated GO terms include "cilium movement," "cilium organization," and "cell projection" (Fig 2B). Some of the up-regulated GO terms were related to mRNA processing and macromolecule localization. However, we were also interested to see that some terms related to mitosis and the cell cycle could be found in the up-regulated GO terms, including "mitotic cell cycle" and "microtubule cell cytoskeleton organization involved in mitosis" (Fig 2A). Overall, this seems to suggest that KMT5B knockdown favours proliferation over formation of cilia.

### Ciliogenic regulators are unaffected by KMT5B knockdown

What could explain this striking down-regulation of ciliogenic genes? We reasoned that knockdown of KMT5B could lead to the down-regulation of a key ciliogenic transcription factor, subsequently misregulating the rest of the ciliogenic differentiation pathway. Of particular interest are members of the E2F transcription factor family, some of which are known to play a role in multiciliogenesis. E2F transcription factors have a variety of distinct functions despite the fact that they share high

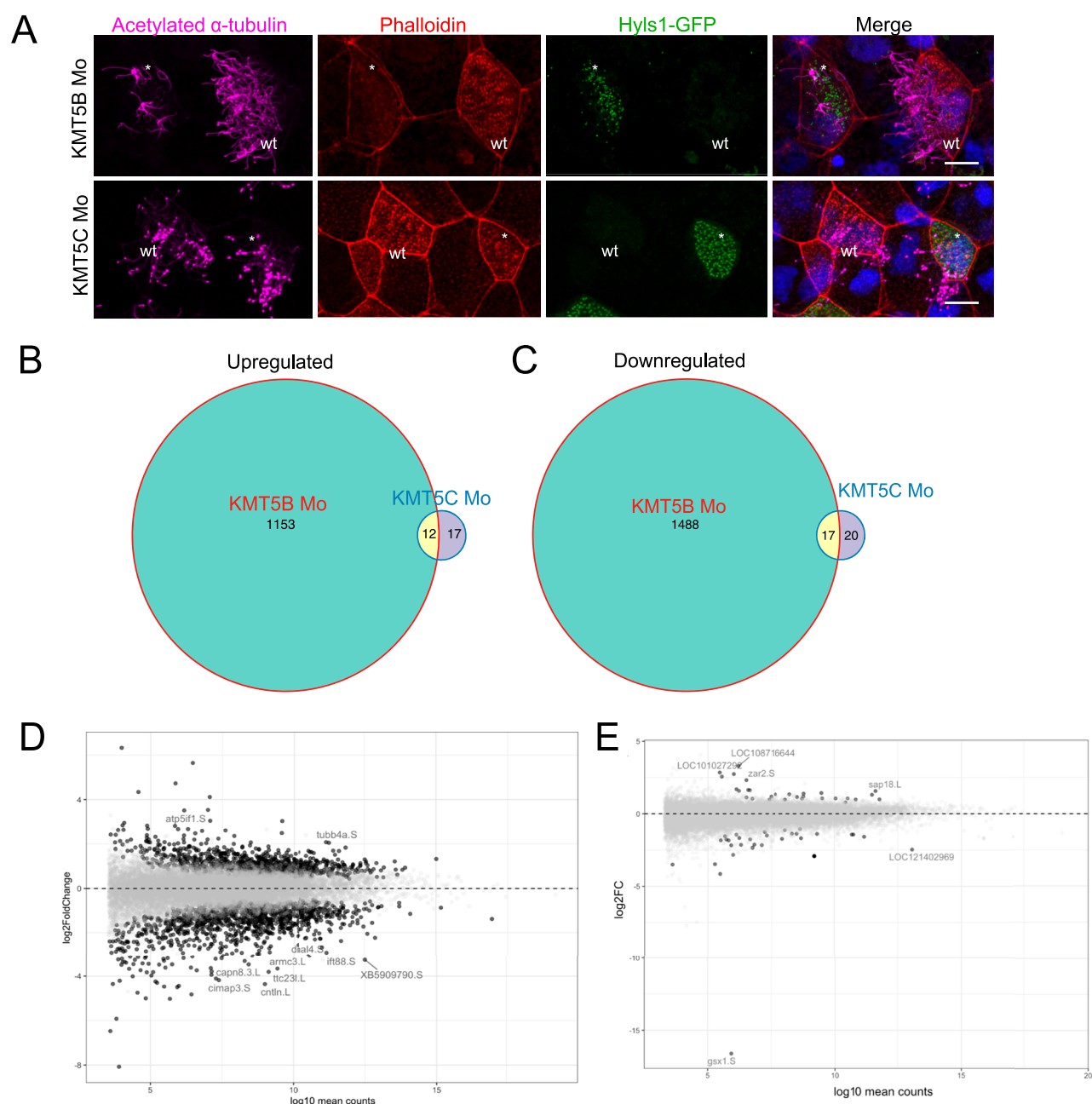

**Figure 1. Knockdown of KMT5B, but not KMT5C, leads to phenotypic and transcriptional changes.**
**(A)** 4-channel confocal images depicting single knockdowns of KMT5B or KMT5C in *Xenopus laevis* embryos. Cilia are magenta (acetylated *α*-tubulin), basal bodies are green (hyls1-GFP), actin meshwork and cell borders are red (phalloidin), and DNA is blue (DAPI). Embryos are injected in one of two ventroanimal blastomeres at the eight-cell stage, giving rise to mosaic embryos in which uninjected (labelled wt in white) and knockdown (white asterisks) multiciliated cells can be visualized in the same field of view. **(B, C)** Venn diagrams showing the number of genes that are up-regulated (B) or down-regulated (C) upon knockdown of KMT5B or KMT5C in *Xenopus laevis* animal caps by RNA-seq compared with control morphants. **(D, E)** MA plot showing $\log_2$ fold change (x-axis) and $\log_{10}$ mean counts (y-axis) in KMT5B (D) or KMT5C (E) knockdown animal caps by RNA-seq.

sequence homology. They can be split into cell cycle–activating and cell cycle–repressing E2Fs. The cell cycle–activating E2Fs are E2F1, E2F2, and E2F3, whereas the cell cycle–repressing E2Fs are E2F4-8. Multicilin, the master regulator of ciliogenesis, directly binds E2F4/5, and this binding is required for basal body biogenesis (Quigley & Kintner, 2017). Recently, E2F7 has been

identified to play a key role in multiciliogenesis by attenuating DNA replication levels to favour MCC differentiation over proliferation (Choksi et al, 2024).

We looked at the change of expression levels of E2F transcription factors and their binding partners (Fig 2D). Both the .L and .S homoeolog of E2F1 were significantly down-regulated. E2F1 is

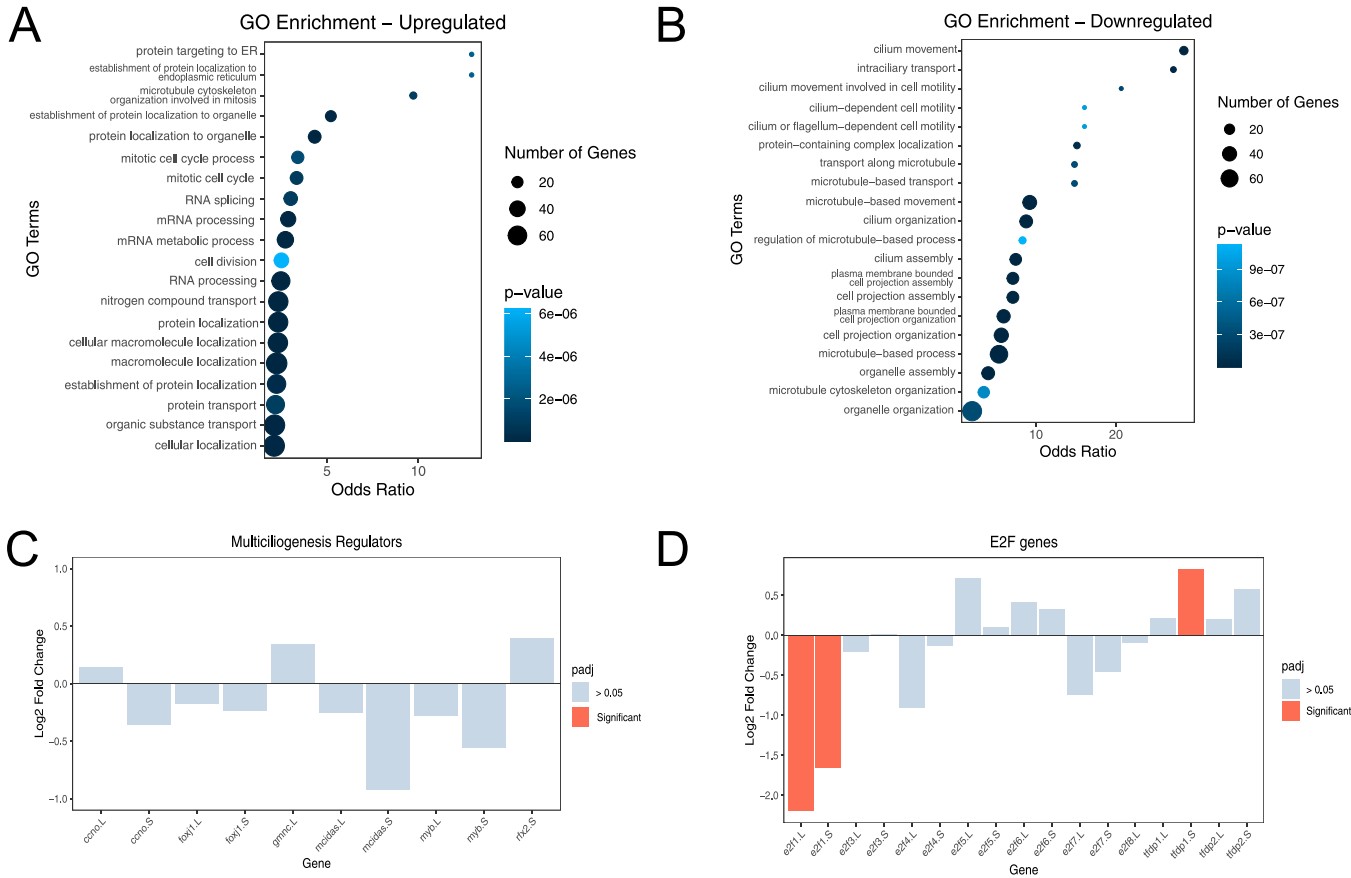

**Figure 2. KMT5B regulates ciliogenesis through an alternative pathway.**
**(A, B)** GO analysis of genes that are up-regulated (A) or down-regulated (B) upon KMT5B knockdown. Up-regulated genes largely relate to mRNA processing and macromolecule localization, whereas down-regulated genes relate to cilia and microtubules. **(C)** Log$_2$ fold change of multiciliogenic regulators upon KMT5B knockdown. None of the major known ciliogenic regulators are significantly misregulated based on RNA-seq results. **(D)** Log$_2$ fold change of E2F genes upon KMT5B knockdown. Only the L and S copies of cell cycle–related e2f1 are down-regulated, whereas tfdp1.S is up-regulated.

primarily involved in cell cycle regulation and has not been shown to play roles in multiciliogenesis. In addition, tfdp1, binding partner of the cell cycle–activating E2Fs, was significantly up-regulated, with a log$_2$ fold change of 0.82. The mRNA levels of other ciliogenic transcription factors like foxj1, rfx2, or myb were not significantly changed (Fig 2C). Taken together, the fact that neither the regulators of ciliogenesis nor most of the E2F transcription factors are misregulated suggests that KMT5B is operating through a different gene expression pathway.

## Catalytic activity of PHF8 rescues ciliogenic phenotype

We previously showed that ciliogenic defects can be partially rescued by PHF8 overexpression, but wanted to determine whether this rescue depends on the catalytic activity. To test this, we obtained two hyperactive, truncated human PHF8 clones (amino acids 1–489), including the catalytic domain (Fortschegger et al, 2010). We injected either wt cDNA (489 wt) or catalytically inactive (489 c.i.) phf8 mRNA into one of two blastomeres of a two-cell-stage embryo alongside LacZ mRNA to act as a lineage tracer. We fixed the embryos at the tadpole stage and performed whole-mount immunocytochemistry against acetylated α-tubulin and performed β-galactosidase staining, which stains LacZ-positive regions blue. We also injected luciferase mRNA to act as a non-specific mRNA control. In each condition, the mRNA was co-injected with either control morpholino or KMT5B morpholino (Fig 3A). Embryos were classified by visual inspection based on the prevalence of acetylated alpha-tubulin staining, prevalence, and size of ciliary tufts, comparing the uninjected and injected sides of the embryo. In total, 95.6% of KMT5B MOs and 86.6% of KMT5B MOs + 489 c.i. PHF8 embryos showed ciliary tuft defects, whereas only 36% of KMT5B MOs + 489 wt PHF8 embryos showed ciliary tuft defects. This indicates that the ciliogenic rescue depends on enzymatic activity of PHF8.

In addition, we noted that both truncated clones affected the spacing of MCCs in wild-type embryos. Ciliary tuft density increased in 81% of 489 wt embryos and 65% of 489 c.i. embryos co-injected with CoMo (Fig 3B). This result, which is independent of PHF8 enzymatic activity, suggests an involvement of PHF8 in MCC specification, rather than MCC differentiation. Further experiments are needed to corroborate this observation.

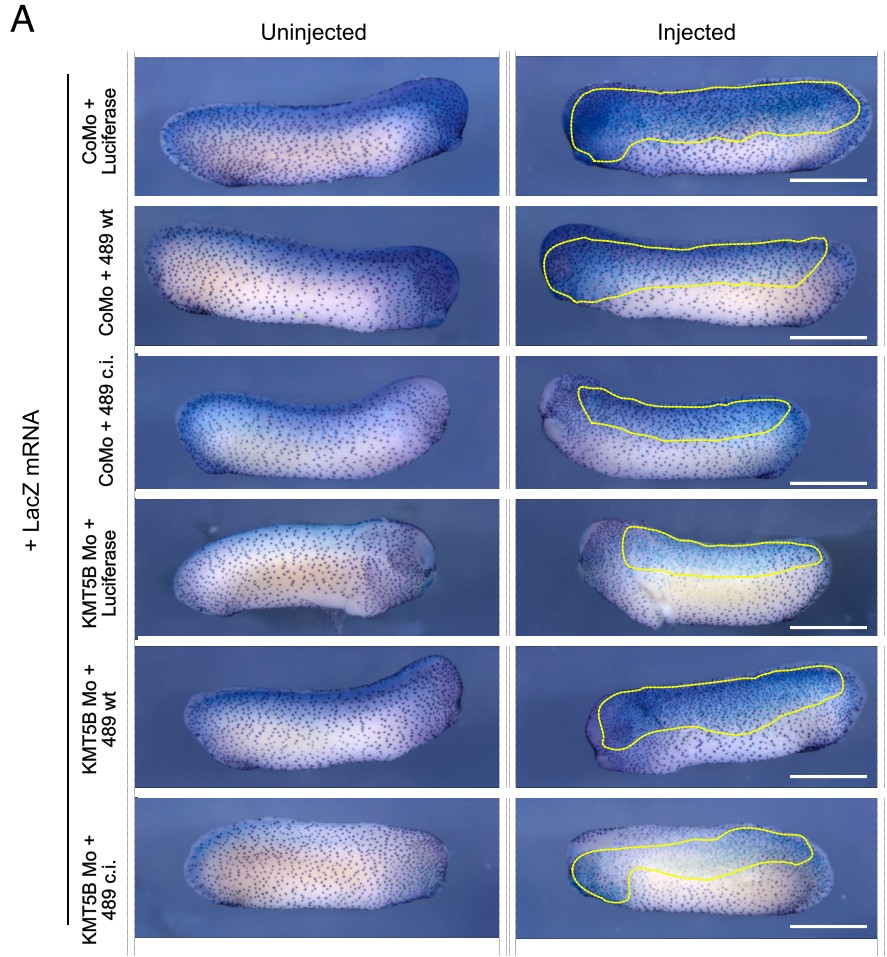

**Figure 3. Catalytic activity of PHF8 rescues the ciliogenic phenotype.**
**(A)** Representative immunocytochemistry images from tailbud-stage embryos injected with KMT5B Mo or CoMo and mRNA luciferase or human PHF8 variants (489 wt/ 489 c.i.). Scale bars = 1 mm (whole embryo) or 200 µm (inserts) and n = 3 biological replicates. **(B)** Quantification of (A). Percentage of embryos either affected by morpholino injections (H1Mo) or rescued by PHF8 mRNA injection. Wt PHF8 mRNA (489 Wt) rescues significantly, whereas catalytically inactive PHF8 mRNA (489 c.i.) shows no significant difference when compared to H1Mo. Significance is indicated by asterisks; *$P < 0.05$, **$P < 0.01$, ns, not significant. N = 4 biological replicates; the yellow dashed line represents regions of ß-galactosidase staining, indicating injected regions.

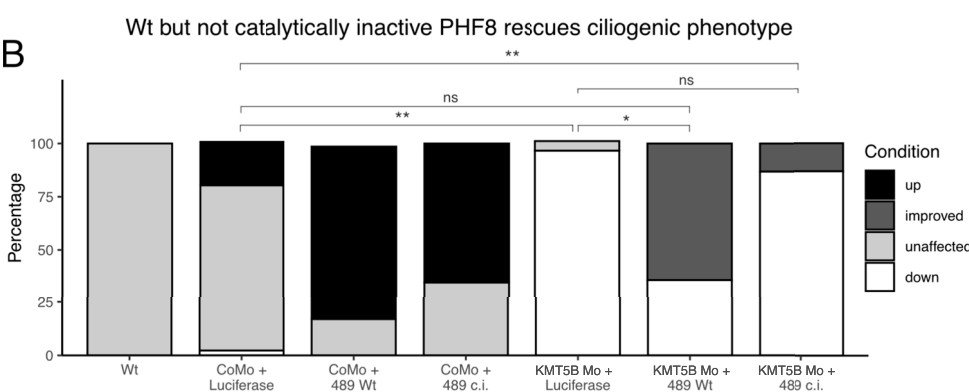

## Multicilin overexpression results in more MCCs, but does not rescue KMT5B phenotype

We wanted to further establish that KMT5B operates outside of the multicilin-driven gene expression pathway, so we decided to deplete KMT5B in the context of multicilin overexpression. Multicilin overexpression has also been shown to induce other epidermal cells to adopt an MCC-like fate (Quigley & Kintner, 2017). In a typical *Xenopus* epidermis, MCCs are one of five cell types,

representing ~20% of cells (Plouhinec et al, 2017). Multicilin overexpression has the added benefit of driving more cells to become multiciliated, giving a more representative profile of MCCs. We used a hormone-inducible multicilin construct containing the ligand binding domain of the human glucocorticoid receptor (MCI-hGR) that has been previously validated (Quigley & Kintner, 2017). To test the effect of KMT5B in multicilin-overexpressing embryos, we injected 60 pg of MCI-hGR into one ventroanimal blastomere of an eight-cell-stage embryos. We co-inject hyls1-GFP to act as a

lineage tracer, indicating which MCCs are wt and which have been affected by multicilin overexpression with or without KMT5B knockdown. At the early gastrula stage (NF11), we induced the embryos using 10 µl dexamethasone. Then, at the tailbud stage, we fixed the embryos and performed whole-mount immunocyto-chemistry against acetylated alpha-tubulin. We see that in embryos injected with control morpholino + MCI-hGR, almost the entire surface of the embryo is covered with MCCs. In contrast, in KMT5B morpholino + MCI-hGR–injected cells appear to adopt a multiciliogenic fate—evident by the multiplied GFP-tagged basal bodies, but they are not able to form cilia (Fig 4A). Because of this finding, we hypothesize that this is truly a phenotype relating to ciliary formation, rather than a cell specification phenotype. These cells can still undergo early stages of multiciliogenesis like amplification of centrioles and emergence into the surface of the embryo, but they cannot form cilia.

We wanted to understand what transcriptional changes were underlying this phenotype, so we performed RNA-seq in animal caps. We injected embryos at the two-cell stage with MCI-hGR and the morpholino of interest. At the blastula stage (NF9), we dissected animal caps, which we then induced with dexamethasone at the gastrula stage (NF11). We harvested the embryos at the neurula stage (NF16) and performed RNA-seq on pools of 10 animal caps. Alongside this, we performed quality control ICCs in embryos to make sure that both the KMT5B knockdown phenotype and the multicilin overexpression phenotype were present in the batches of animal caps selected for sequencing. We also staged animal caps based on these sibling embryos. (For a schematic of the MCI-induction method, including the RNA-seq workflow, see Fig S3A; for representative images and quantification of quality control embryos, see Fig S3B and C).

The overexpression of MCI with control morpholino injection (CoMo + MCI-hGR) led to the misregulation of 591 genes (Fig S4A and B). Of this, 525 genes were up-regulated and only 66 genes were down-regulated. In line with our expectations, GO analysis of the up-regulated gene categories revealed an enrichment in categories related to MCCs was up-regulated including "centriole assembly," "cilium assembly," and "cilium organization" (Fig S4C). On the other hand, the down-regulated genes were enriched for GO categories related to the cell cycle including "DNA replication" and "DNA replication initiation," as well as a category related to a different cell lineage, "mesoderm development" and "negative regulation of cell differentiation" (Fig S4D). The down-regulation of genes related to negative regulation of cell differentiation is unsurprising because MCCs are post-mitotic and terminally differentiated, and so DNA replication and its related processes should be restricted.

## Transcriptional changes are driven by KMT5B knockdown

MCI-hGR overexpression results in up-regulation of ciliogenic genes, whereas KMT5B knockdown results in down-regulation of ciliogenic genes. We wondered whether concurrent MCI-hGR overexpression and KMT5B depletion would have competing effects on ciliogenic gene expression. We found KMT5B knockdown to dominate over MCI-GR activity (Fig 4A). To better understand how these regulators affect the ciliogenic gene expression programme,

we performed a combined analysis of the two RNA-seq experiments, Group A (independent knockdowns of KMT5B and KMT5C) and Group B (MCI-hGR overexpression experiment). We found two major clusters, both defined by the behaviour of genes in response to KMT5B knockdown (Fig 4B): one cluster that is defined by genes that are down-regulated in KMT5B knockdown and KMT5B knockdown + MCI-hGR, and a second cluster that is defined by genes that are up-regulated in KMT5B knockdown and KMT5B knockdown + MCI-hGR. This indicates that KMT5B knockdown, and not MCI-hGR overexpression, is the driver of these transcriptional changes.

We also performed GO analysis on KMT5B knockdown + MCI-hGR–overexpressing animal caps. In the up-regulated genes, we see categories relating to RNA processing and a number of metabolic processes; however, unlike in the KMT5B knockdown, there is no cell cycle connotation (Fig 4C and D). The down-regulated genes are once again dominated by ciliogenic GO terms. This is despite the fact that ciliogenic genes are up-regulated in MCI-hGR–overexpressing animal caps, and the fact that MCI-hGR overexpression results in a higher density of MCCs (Fig 4C and E).

Because KMT5B knockdown and MCI overexpression have opposing effects on the differentiation of MCCs, we decided to explore whether these two conditions are regulating the same subset of multiciliogenic genes. We found that KMT5B knockdown and MCI overexpression regulate partially overlapping but largely distinct gene sets. There is a defined overlap of 154 genes between genes down-regulated in the KMT5B knockdown and genes up-regulated upon MCI overexpression. However, most of the genes are regulated in only one of the two contrasts and do not show coordinated regulation across conditions (Fig S5A).

We then performed GO analysis for biological processes on the overlapping genes between genes up-regulated in MCI overexpression and down-regulated in KMT5B knockdown and found that the most enriched categories were "intraciliary transport," "cilium movement," and "axoneme assembly." These top categories are highly similar to the top down-regulated categories in KMT5B knockdown overall; however, the most enriched up-regulated GO categories for MCI-hGR overexpression overall are "centriole replication," "centriole assembly," and "centrosome duplication" (Fig S5B).

These results argue against a simple linear model in which MCI overexpression rescues or replaces the transcriptional effects of KMT5B loss. Instead, they indicate that KMT5B and MCI act on overlapping but non-identical gene sets, with coordinated, opposite regulation confined to a limited subset of shared targets. Taken together, this suggests that regulation of multiciliogenesis by KMT5B proceeds either through an alternative pathway, or downstream of the MCI-initiated transcription factor cascade.

Based on these results, one might be tempted to speculate that KMT5B acts simply downstream of MCI. For instance, it might enable FOXJ1 to form single motile cilia. For MCCs, we showed previously that overexpressed FOXJ1 did not rescue ciliary axonemes in the KMT5B/C knockdown condition (Angerilli et al, 2023). Therefore, we have started to investigate gastrocoel roof plate (GRP) cilia (Fig S6), which are required to establish L/R asymmetry in the embryo (reviewed by Blum et al [2014]). GRP cells express FOXJ1, but not MCI, and form single, motile cilia, whose stroke

**Figure 4. KMT5B knockdown drives transcriptional and phenotypic changes even in the presence of MCI-hGR.**
**(A)** Four-channel confocal microscopy images showing *Xenopus laevis* embryos injected with control morpholino (CoMo) + MCI-hGR or KMT5B morpholino + MCI-hGR. Cilia are magenta (acetylated α-tubulin), basal bodies are green (hyls-GFP), actin meshwork and cell borders are red (phalloidin), and DNA is blue (DAPI). Embryos are injected in one of two ventroanimal blastomeres at the eight-cell stage, giving rise to mosaic embryos in which wt and knockdown multiciliated cells can be visualized in

generates directional fluid flow. We injected CoMo or KMT5B-Mo into the dorso-marginal zone of the embryo and derived from this region open-face explants, which we cultured until the initial neural tube stage, when GRP cilia are fully developed. At this time point, differentiation of epidermal MCC is under the influence of KMT5B. We recorded scanning electron micrographs of the inner surface and analysed GRP ciliary length and number (Fig S6A–D), following an established protocol (Walentek et al, 2013). Knock-down of KMT5B had no significant effect on the formation of single, motile cilia in these cells (Fig S6E and F). These results suggest that cilia-promoting activity of KMT5B could be coupled specifically to MCI-controlled multiciliogenesis, but having no effect on FOXJ1-dependent formation of single motile cilia.

### Effect of KMT5B on chromatin accessibility

H4K20me1 has been shown to have roles in chromatin compaction through its reader protein L3MBTL1, and through interactions with the C terminus of H2A on neighbouring histone tails. Could the down-regulation of ciliogenic genes result from a loss of chromatin accessibility at target genes? We decided to use ATAC-seq to address this question. ATAC-seq peaks indicate open regions of the chromatin, and novel or changing peaks indicate differential accessibility.

We performed ATAC-seq in control morphant and KMT5B morphant animal caps in triplicate across three biological replicates. We identified 118,809 total peaks, a comparable number to previous ATAC-seq experiments in Xenopus animal caps (Esmaeili et al, 2020) (Fig 5A). In total, 24,604 (20.7%) are found at promoters, 1,648 (1.4%) immediately downstream of the TSS, 1,512 (1.3%) at the 5′ UTRs, 996 (0.8%) at the 3′ UTRs, 4,489 (3.8%) in the exons, 42,188 (36%) in the introns, and 43,372 (36%) in the intergenic regions. Of these peaks, 556 are differentially accessible (Fig 5B), representing 0.05% of all peaks. Most of these peaks were increasing in accessibility, whereas only 17 peaks became less accessible. Upon manual inspection of the local browser tracks, the less accessible peaks seemed to largely be called as the result of higher peaks in one of three control morphant-injected tracks and may not have biological relevance. Contrary to our expectations, we did not find chromatin accessibility to change significantly on ciliogenic genes. When we look at three of the most misregulated genes from the KMT5B knockdown RNA-seq experiment, ttc23.L, dnal4.S, and ift88, their accessibility does not change (Fig 5C). The most significantly changing peaks are found mostly in intergenic regions (Fig 5D) and are generally increasing in accessibility rather than decreasing. We also considered key ciliogenic transcription factors and found no difference in accessibility (Fig S7). Overall, this does not support the hypothesis that KMT5B exerts its control on ciliogenic gene expression by affecting chromatin accessibility in a global manner. However, this analysis cannot rule out the possibility that subtle, yet functionally critical, chromatin alterations at specific enhancers, insulators, or other regulatory elements could cause the observed ciliogenic defect in MCCs.

## Discussion

MCCs are a highly specialized and terminally differentiated cell type, which represent a critical component of mucociliary epithelia found on the human alveoli and the embryonic skin of *Xenopus* (Stubbs et al, 2006; Werner & Mitchell, 2012; Walentek, 2021). Multiciliogenesis is a complicated cellular differentiation process by which epidermal cells generate hundreds of motile cilia, and this is executed through a complex gene expression programme (Stubbs et al, 2012; Meunier & Azimzadeh, 2016). Here, we report that the H4K20 methyltransferase KMT5B regulates Xenopus multiciliogenesis in the larval epidermis, whereas single, motile cilia present on the gastrocoel roof plate appear unaffected. We further show that KMT5B knockdown in animal caps leads to the concerted down-regulation of ciliogenic genes, even when the master regulator of multiciliogenesis MCI is overexpressed. The phenotypic link between KMT5B knockdown and MCI over-expression is restricted, given that the two proteins act on over-lapping, but non-identical, gene sets.

There are several mechanisms, by which KMT5B, but not KMT5C, could promote ciliogenesis in MCCs. We discuss these below and relate them to our experimental data presented here and in Angerilli et al (2023). First, the enzyme may regulate ciliary gene transcription by converting H4K20me1 to H4K20me2, which is clearly perturbed by KMT5B knockdown. In support of this hy-pothesis, we have previously shown that the ciliogenic phenotype is specific and dependent on the catalytic activity of KMT5B. We generated the phenotype using two non-overlapping morpholinos and demonstrated that only catalytically active KMT5B mRNA is able to rescue (Angerilli et al, 2023).

It may seem surprising that knockdown of KMT5C has no effect on ciliary tufts, although it impacts H4K20me1/me2 levels to a similar extent as KMT5B (Angerilli et al, 2023). However, there is clear evidence that the two genes are not fully redundant and their functions have partly diverged. For example, only KMT5C catalyses H4K20me3 (Angerilli et al, 2023); the kmt5b gene is essential in mice, but kmt5c is not (Schotta et al, 2008); amino acid sequences of the catalytic domains are higher conserved between frog and murine homologues (aa identity 98% for KMT5B, 88% for KMT5C) than between the two murine genes (75%). Finally and potentially relevant for our study, they are known to use different C-terminal sequences to target heterochromatin; FRAP experiments have shown that KMT5C interacts stably with heterochromatin, whereas KMT5B interacts much more transiently (Hahn et al, 2013). KMT5C may therefore be largely trapped within heterochromatin, whereas

the same field of view. Uninjected cells are labelled in white as "wt," whereas injected cells are indicated with an asterisk (*). Scale bars = 10 $\mu$m. **(B)** Cluster analysis heatmap of top responding genes across all conditions and replicates. Genes group into two main clusters: genes down-regulated or up-regulated upon KMT5B knockdown (in the absence or presence of MCI-hGR). Group A includes replicates from the single knockdown RNA-seq experiment, and Group B includes replicates from the MCI-hGR containing RNA-seq experiment, and samples are batch-corrected between experiments. **(C)** MA plot showing all genes (light grey), significantly misregulated genes (dark grey), ciliary genes (yellow), and significantly misregulated genes (red). **(D, E)** GO analysis showing the up-regulated (D) and down-regulated (E) gene categories upon MCI-hGR overexpression. The down-regulated gene categories are largely related to ciliogenesis and microtubule assembly.

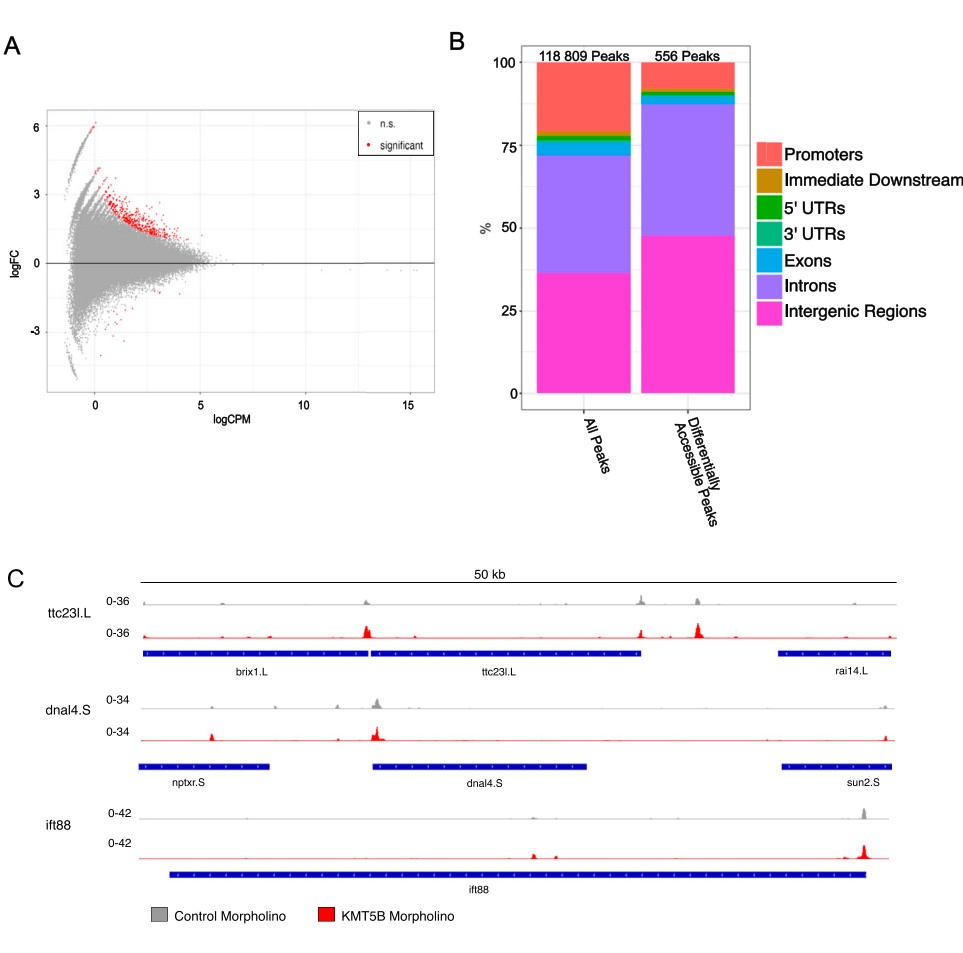

**Figure 5. Depletion of KMT5B leads to increased chromatin accessibility.**
**(A)** MA plot showing the distribution of peaks in KMT5B knockdown animal caps. Significantly changing peaks are shown in red ($P < 0.05$). Most of the peaks are increasing rather than decreasing, indicating a mild chromatin opening effect of KMT5B knockdown in *Xenopus laevis* animal caps.
**(B)** Distribution of peaks across genomic features in all peaks and changing peaks. N = 3 biol. replicates. **(C)** Representative browser tracks of the most significantly down-regulated ciliogenic genes from the KMT5B knockdown RNA-seq experiment.
**(D)** Representative browser tracks of the most significantly changing peaks between the KMT5B knockdown and control animal caps.

KMT5B would act on both eu- and heterochromatin. Such a difference could provide a spatial segregation of the two enzyme activities within chromatin that may ultimately determine their different target genes and their different impacts on transcriptional regulation. Until specific antibodies against KMT5B/C become available for ChIP analysis, this hypothesis cannot be directly addressed.

KMT5B knockdown leads to a profound shift from H4K20me2 to H4K20me. That this shift is important for the ciliogenic phenotype is supported from rescue experiments with PHF8, a Jumonji C (JmjC) domain containing histone demethylase. It is targeted to promoters by binding of H3K4me3 through its PHD finger, and the catalytic activity is carried out by the JmjC domain. PHF8 knockdown also leads to a number of craniofacial and cytoskeletal defects. In PHF8-depleted zebrafish embryos, the pharyngeal arches are reduced or absent, and brain development is impaired (Qi et al, 2010). PHF8 is also involved in regulating cell cycle progression (Liu et al, 2010). Originally, we found that the overexpression of both full-length human PHF8 protein and a truncated *Xenopus tropicalis* PHF8 variant partially restored ciliary tufts in the KMT5B knockdown condition (Angerilli et al, 2023). Work by others has demonstrated that catalytic activity requires both JmjC and JmjD domains, corresponding to a minimal length of 447 amino acids (Fortschegger et al, 2010; Loenarz et al, 2011). Because our Xenopus PHF8 clone encodes a much shorter protein, we considered its activity at least severely compromised. We therefore speculated that full-length human PHF8 achieves ciliogenic rescue by demethylating H4K20me1, whereas the Xenopus variant might improve ciliogenesis by masking H4K20me1 marks (Angerilli et al, 2023). Unfortunately, we could not detect PHF8-

dependent changes of H4K20me0/H4K20me1 levels by quantitative mass spec analysis to validate this assumption, most likely because the local activity of PHF8, which acts primarily on promoters, is masked within the huge excess of H4K20me1, arising from the absence of KMT5B.

We therefore decided to readdress this point using a pair of nearly identical human PHF8 protein variants (i.e., PHF8 [1–489] NLS) that are either wild-type, active, or double-mutated at H247/D249 to alanine, being enzymatically inactive (a kind gift of Dr. Ramin Shiekattar). These constructs unambiguously showed that only enzymatically active PHF8 restores ciliary tufts in KMT5B-depleted organoids. In hindsight, this result implies that the truncated Xenopus PHF8 protein has some residual demethylation activity, although this needs to be shown. In summary, these rescue experiments support the hypothesis that excess H4K20me1 is detrimental to the ciliogenic transcription programme and removal of this mark by PHF8 can partly alleviate this effect. Alternatively, PHF8 has originally been characterized to demethylate histone H3 at several lysine residues (H3K9me1/me2, H3K27me1/me2, H3K36me2; see Fortschegger et al, 2010; Loenarz et al, 2011); such events could promote a more favourable chromatin environment without acting on H4K20me1. However, we have not detected changes in histone H3 methylation in human PHF8-overexpressing Xenopus embryos (Angerilli et al, 2023).

It needs to be mentioned that H4K20me1 has an ambiguous role in transcription. In some contexts, it seems to be a transcriptional activator (Barski et al, 2007; Shoaib et al, 2021), whereas in other contexts, it is a transcriptional repressor (Trojer et al, 2007; Congdon et al, 2010; Gurvich et al, 2010). The mark is largely found in the gene bodies of active genes. ChIP and ATAC-seq data also correlate H4K20me1 to the transcription of short house-keeping genes (Shoaib et al, 2021). However, SET8 depletion results in a twofold increase in the expression of H4K20me1-decorated genes (Congdon et al, 2010). H4K20me1 reader L3MBTL1 has also been linked to transcriptional repression, and demethylation of H4K20me1 by PHF8 derepresses a subset of E2F1-regulated promoters (Boccuni et al, 2003; Trojer et al, 2007; Gurvich et al, 2010). Altogether, the role of H4K20me1 in transcription is unclear, and may be context- or location-specific.

Alternatively, the ciliogenesis phenotype may not involve chromatin structure. Many histone-modifying enzymes have non-catalytic functions or may methylate non-histone substrates. Because catalytic activity of both KMT5B and PHF8 is required for regulating ciliogenesis, it could be that both of these enzymes are methylating non-histone targets. KMT5B has previously been shown to methylate in vitro non-histone proteins including CASZ and OSBPL1A (Weirich et al, 2016), and SET8 has been shown to methylate $\alpha$-tubulin (Chin et al, 2020). Non-histone substrates have not yet been identified for PHF8. Furthermore, recent data from Drosophila with H4K20 histone mutants suggest that H4K20 writer and reader proteins SET8 and L(3)MBT may not require H4K20me to carry out their functions (Crain et al, 2024). This could also be the case for other H4K20-modifying enzymes like KMT5B.

We previously demonstrated that KMT5B knockdown cannot be rescued by the overexpression of MCI (Angerilli et al, 2023). In wild-type embryos, multicilin overexpression is sufficient to induce multiciliogenesis in non-MCCs. The promotion of MCC specification by multicilin is maintained in KMT5B knockdown embryos, based

on the unambiguous presence of multiplied basal bodies; however, these cells still lack ciliary tufts. On the transcriptional level, the overexpression of MCI in control morphant animal caps led to a significant up-regulation of ciliogenic genes, whereas these genes were significantly down-regulated when multicilin was overexpressed in KMT5B knockdown caps. The changes in transcription are largely driven by KMT5B knockdown, not multicilin overexpression, despite the fact that multicilin overexpression leads to a dramatic remodelling of the embryonic epidermis. Taken together, this suggests that regulation of multiciliogenesis by KMT5B is achieved through an alternative pathway to the canonical multiciliogenic programme.

But how is this regulation achieved? Because mapping H4K20me1 by ChIP-seq was not technically feasible, we have investigated whether KMT5B alters chromatin accessibility, either at ciliogenic genes or as a whole. However, profiling accessibility by ATAC-seq showed only mild effects on the chromatin. Only 0.05% of peaks were affected, and they tended to actually increase in accessibility rather than decrease. Accessibility in the regulatory regions of ciliogenic genes was not affected. In fact, very few differentially accessible peaks were mapped to genes, and were instead found in intergenic and intronic regions. Overall, this suggests that KMT5B does not exert its regulatory role through alteration of chromatin accessibility (Barski et al, 2007; Shoaib et al, 2021).

We previously showed that KMT5B is responsible for writing ~50% of H4K20me2, whereas the remaining 50% is written by KMT5C (Angerilli et al, 2023). Following up on this finding, new data here indicate that mono- or dimethylation of H4K20 has little effect on chromatin accessibility in a global or significant manner. Regarding the function of KMT5B in ciliogenesis, it is still unclear whether this process is regulated through methylation of H4K20 or through an additional function of the enzyme itself. Our findings may suggest the presence of an unknown reader of H4K20me2 that drives ciliogenesis through deposition of the mark. Alternatively, KMT5B may carry out its role in ciliogenesis separate from H4K20 methylation, possibly through the methylation of a yet-unknown non-histone target. KMT5B is an essential gene, with knockout leading to perinatal lethality in mice (Schotta et al, 2008). Further research is required to establish the mechanism underlying these essential functions, including its role in ciliogenesis.

# Materials and Methods

## Ethics statement

Xenopus laevis and X. tropicalis were acquired from Nasco and Xenopus1. Xenopus experiments adhere to the Protocol on the Protection and Welfare of Animals and are approved by the local Animal Care Authorities (license number: 03-22-042).

## Expression constructs and morpholino oligonucleotides

Translation-blocking morpholino oligonucleotides directed against KMT5B (X. laevis KMT5B 5′-ggattcgcccaaccacttcatgcca-3′), KMT5C

(*X. laevis* KMT5B: 5'-ttgccgtcaaccgatttgaacccat-3'), and standard control morpholino (5'-cctcttacctcagttacaatttata-3') were obtained from Gene Tools LLC. In total, 30–40 ng of morpholino was injected per blastomere into two-cell-stage *X. laevis* embryos. For confocal analysis, embryos were injected at the eight-cell stage with 5 ng of morpholino into the ventroanimal blastomere.

Rescue experiments were performed with wild-type truncated human PHF8 (1–489) and catalytically inactive truncated human PHF8 (1–489) in pCS2+, kindly provided by R. Shiekhattar (Fortschegger et al, 2010) and MCI-hGR in pCS2+, kindly provided by P. Walentek. Furthermore, we injected Hyls1-GFP in pCS2+ kindly provided by A. Dammerman, and LacZ in pCS2+ as lineage tracers. Lastly, we injected luciferase mRNA as a control. Synthetic mRNAs were injected at the two-cell and eight-cell stage.

### *Xenopus* methods

*X. laevis* adults and embryos were handled using standard protocols as described in Showell & Conlon (2009). Two-cell-stage embryos were injected with up to 5 nl per blastomere, and volume was scaled based on the embryonic stage. Eight-cell-stage embryos were injected with 1.25 nl per blastomere. Two-cell-stage embryos were co-injected with Alexa 488 dextran (Invitrogen), and fluorescently stained embryos were co-injected with Hyls1-GFP mRNA, both as lineage tracers. Two-cell-stage embryos were sorted left- and right-injected based on fluorescence. Embryos were cultured in 0.1× MMR at temperatures ranging from 16° to 23° and staged based on the Nieuwkoop–Faber table of *Xenopus* development (NF1967).

Animal caps were manually dissected by excising the blastocoel roof of blastula-stage embryos (NF9). Animal caps were then transferred to a petri dish with individual wells to avoid amalgamation. The animal caps were incubated in Steinberg's solution. MCI-hGR–injected animal caps and embryos were induced at the midgastrula stage (N11) with 10 $\mu$M dexamethasone (Sigma-Aldrich).

### Immunocytochemistry and confocal microscopy

Whole-mount immunocytochemistry was performed as previously described (Robinson & Guille, 1999) on tailbud-stage embryos (NF28). Confocal stainings were performed the same way, with the exception of skipping the methanol step and permeabilizing the cell membranes with a 20-min incubation in PBS with 2% Triton. Embryos were incubated with monoclonal anti-acetylated $\alpha$-tubulin antibody (T6793 1:500; Sigma-Aldrich) as a primary antibody, and an alkaline phosphatase-conjugated secondary antibody (sheep × mouse Fab IgG Alk Phos, 1:1,000; Chemicon).

For confocal staining, a fluorescent secondary antibody (goat anti-mouse 1:1,000; Chemicon) was used. Nuclei were stained by incubating in DAPI (1:50 in PBS; Sigma-Aldrich) for 15 min, and cell borders and the apical actin meshwork were stained in 5% Alexa Fluor 555 Phalloidin (Cell Signaling) in PBS for 1.5 h. Embryos were mounted between two glass coverslips separated by 0.35-mm-thick double-sided wells cut into the tape and filled with modified DABCO mounting solution (85.43% glycerol, 10% 10× PBS + 2% DABCO, 4.57% $H_2O$).

Confocal microscopy was performed at the Core Facility Bioimaging of the Biomedical Center with an inverted Leica SP8X WLL microscope, equipped with 405-nm, argon and white light laser (470–670 nm) and acousto-optical beam splitter.

Image stacks were acquired with a 93×/1.3 NA glycerol immersion objective with motorized correction collar to adjust for refractive index mismatches. Images were recorded at zoom 1 with a format of 1,640 × 1,640 pixels at 200 Hz unidirectional scan speed. Image pixel size was 76 nm, z-step size was 332 nm, and the pinhole was set to 1.0 AU (580-nm reference wavelength).

The following fluorescence settings were used: DAPI (excitation 405 nm; emission 419–474 nm), Hyls1-GFP (argon 488 nm; 494–521 nm), Cy3 (553 nm; 582–601 nm), and Alexa Fluor 647 (652 nm; 677–697 nm). Recording was done stack sequential with two sequences: Seq 1: GFP, Cy3, Alexa Fluor 647; Seq 2: DAPI. Signals were recorded with hybrid photodetectors (HyDs) in standard mode, and only DAPI was recorded with conventional photomultiplier tubes.

Raw image stacks were deconvolved with Huygens Professional 17.04 (Scientific Volume Imaging), exported to Leica LAS × 3.5.7 with histogram normalization set to contrast stretch, and saved in lif format.

### GRP cilia and scanning electron microscopy

Gastrocoel roof plate cilia were analysed by targeted microinjection into the dorsal marginal zone of two-cell embryos, next to the newly formed cleavage plane. The prospective dorsal and ventral sides of the embryo can be often distinguished by asymmetric pigment distribution, where the dorsal side is less pigmented. Because lineage tracing by fluorescent dextrans turned out to interfere with the SEM sample preparation, injections were restricted exclusively to embryos with unambiguous pigment asymmetry. Training injections with fluorescent dextrans ensured that the gastrocoel roof plate area was consistently targeted. Afterwards, CoMo and KMT5B-Mo were injected at the same concentration as for animal cap injections. GRP explants were microdissected at NF17 (late neural fold stage) and fixed immediately in Sorensen's phosphate buffer with 3% glutaraldehyde for 1 h to prevent explants from closing up. Subsequently, the explants were manually cleaned from cell debris and washed at least 3 × 20 min in PBS, followed by a second fixation in 1% OsO4 and stepwise dehydration into 100% ethanol. All samples were attached on a specimen stub and sputter-coated with gold. SEM images were kindly recorded by Mrs. Beate Aschauer from the department of Prof. Dr. Ulrich Welsch (Lehrstuhl Anatomie II, Institute of Anatomy, LMU Munich) on a JEOL:JSM-35 GF scanning electron microscope. Image orientation was determined in reference to the visible blastopore, marking the posterior end of the explant. An area of 320 × 320 $\mu$m was selected from the centre of the GRP image to determine cell boundaries, ciliary length, and position using ImageJ as described in Walentek et al (2013).

### Statistical analysis

ICC results from knockdown experiments were analysed using a two-tailed $t$ test. Results from rescue experiments were analysed using one-way ANOVA with post hoc Tukey's test. Asterisks indicate $P$-values: $*P < 0.05$, $**P < 0.01$, $***P < 0.001$.

### RNA library preparation and sequencing

Ten *X. laevis* animal caps per condition were harvested and snap-frozen on liquid nitrogen. RNA was extracted using RNeasy Mini Kit with on-column DNase digestion (QIAGEN). Total RNA was measured on a 4200 TapeStation (Agilent), and the quality was checked to ensure a $RIN^e$ of at least 7. Library preparation was performed using NEBNext Ultra II Directional RNA Library Prep Kit for Illumina (New England Biolabs). PolyA(+) mRNA was selected for using the NEBNext Poly(A) Magnetic Isolation Module (New England Biolabs). NEBNext Multiplex Oligos for Illumina(Index Primers Set 1 and 2) were used for sample indexing. Size selection was performed using AMPure XP beads (Beckman Coulter). After preparation, the finished libraries were rerun on a 4200 TapeStation (Agilent) using the HSD1000 ScreenTape (Agilent). Sequencing was performed on an Illumina NextSeq 1000 with 50-bp paired-end reads to a depth of 20 million reads.

### RNA-seq analysis

All data processing methods were applied using default parameters unless specified. Expression quantification was performed using kallisto (version 0.48) using XENLA_10.1 version of the *X. laevis* genome. In R/Bioconductor, expression data were collapsed from isoform to gene level for downstream processing. Differential expression was assessed using DESeq2 (version 1.42.1), using the experimental batch as a random factor, and genes with an adjusted $P < 0.1$ were considered differentially expressed. For gene ontology enrichment analysis, statistical significance was assessed using a hypergeometric framework equivalent to Fisher's exact test (topGO, version 2.62) and ontologies provided by org.Xl.eg.eb (version 3.22).

### ATAC library preparation and sequencing

Two animal caps per condition were cultured until the neurula stage (NF16) and collected for ATAC-seq. Samples were prepared as in Buenrostro et al (2013) and Esmaeili et al (2020). In short, animal caps were centrifuged for 5 min (500$g$, 4°C) in PBS. The PBS was replaced with lysis buffer (10 mM Tris–HCl [pH 7.5], 4 mM MgCl$_2$, 10 mM NaCl, 0.1% Igepal CA-630), and animal caps were homogenized by pipetting. The tagmentation reaction was performed in 25 $\mu$l TDE1 buffer, 1.88 $\mu$l TDE1 enzyme (20034197; Illumina) diluted to a final volume of 50 $\mu$l with nuclease-free H$_2$O at 37°C with 350 rpm (~0.020$g$) shaking for 30 min. Libraries were prepared according to Buenrostro et al (2013) and assessed using a TapeStation (Agilent) before and after bead-based size selection. Samples were sequenced with 50-bp paired-end reads to a depth of 20 million reads per sample on an Illumina NextSeq 1000.

### ATAC-seq analysis

All data processing methods were applied using default parameters unless specified. After adapter trimming with cutadapt, sequencing reads were aligned to the XENLA_10.1 version of the *X. laevis* genome using bowtie2 (version 2.4). Duplicates were marked and removed with Picard Tools. Peaks were called using macs2. Aligned reads were read in R/Bioconductor, converted to coverages (library GenomicRanges), and exported to bigWig files (library rtracklayer). Differential accessibility was assessed using csw/edgeR and TMM normalization as described in the csaw workflow detailed in https://github.com/reskejak/ATAC-seq/blob/master/csaw_workflow.R.

## Data Availability

RNA high-throughput sequencing data have been deposited in the NCBI GEO under the accession number GSE274392. ATAC-seq data have been deposited in the NCBI GEO under the accession number GSE274391.

## Supplementary Information

## Acknowledgements

We thank Drs. Peter Walentek, Alexander Dammerman, and Ramin Shiekhattar for their kind gift of recombinant plasmids; Beate Aschauer and Dr. Ulrich Welsch for SEM image recordings; Drs. Axel Schweickert and Thomas Thumberger for assistance in GRP cilia analysis; Dr. Andreas Thomae and the Core Facility of Bioimaging of the Biomedical Center, LMU Munich, for the instructions and technical support in confocal imaging; and the BMC Core Facility Animal Models (CAM) for animal care and maintenance. This work was funded by the Deutsche Forschungsgemeinschaft (DFG, German Research Foundation)—Project-ID 213249687—SFB 1064 (Project A12).

### Author Contributions

J Tait: conceptualization, data curation, formal analysis, validation, investigation, visualization, methodology, and writing—original draft, review, and editing.
C Marthen: visualization and methodology.
B Hoelscher: methodology.
T Straub: data curation and formal analysis.
O Hsam: data curation, formal analysis, visualization, and methodology.
RAW Rupp: conceptualization, supervision, funding acquisition, project administration, and writing—original draft, review, and editing.

### Conflict of Interest Statement

The authors declare that they have no conflict of interest.

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
