## [Reviewer comments · Life Science Alliance]

Catalytic activity of KMT5B promotes ciliogenesis without affecting global chromatin accessibility

Janet Tait, Carmen Marthen, Barbara Hoelscher, Tobias Straub, Ohnmar Hsam, and Ralph Rupp

DOI: <https://doi.org/10.26508/lsa.202503455>

Corresponding author(s): *Ralph Rupp, Ludwig-Maximilians-Universität München*

Review Timeline:

Submission Date:	2025-07-15
Editorial Decision:	2025-09-08
Revision Received:	2026-02-11
Editorial Decision:	2026-02-27
Revision Received:	2026-03-17
Accepted:	2026-03-19

Scientific Editor: *Tim Fessenden*

Transaction Report:

September 8, 2025

Re: Life Science Alliance manuscript #LSA-2025-03455

Prof. Ralph A.W. Rupp
Ludwig-Maximilians-Universität München
Biomedical Center, Molecular Biology
Grosshaderner Strasse 9
Planegg-Martinsried D-82152
Germany

Dear Dr. Rupp,

Thank you for submitting your Follow Up manuscript entitled "Catalytic activity of KMT5B promotes ciliogenesis without affecting accessibility of ciliary genes" to Life Science Alliance. The manuscript was assessed by expert reviewers, whose comments are appended to this letter. We sincerely regret the long delay faced by this manuscript due to reviewer and editor availability, and we appreciate your patience.

As you will see, reviewers diverged in their enthusiasm for this work which offers new observations related to your prior work published with LSA. We welcome submission of a suitably revised manuscript to LSA. Although we appreciate the reservations of Reviewer 1 on the advance of this work relative to the prior work, this overlap is acceptable for a Follow Up manuscript. Please attend to the distinction noted by this reviewer on rescues with human PHF8 vs truncated *Xenopus* PHF8. Reviewer 2 made several important requests to verify key observations and clarify methodology. In particular, their points related to Fig 1A (on cell size) and Fig 4A (on basal body docking) must be addressed in a revised manuscript. Reviewer 3 sought confirmation of epigenetic marks by KMT5B and PHF8, and while data for the former were shown previously we agree that confirming the effects of PHF8 rescue on histone marks should be included here. Please also discuss the remaining questions posed by this reviewer.

Thank you for this interesting contribution to Life Science Alliance. We are looking forward to receiving your revised manuscript.

Sincerely,

- A letter addressing the reviewers' comments point by point.
- An editable version of the final text (.DOC or .DOCX) is needed for copyediting (no PDFs).
- High-resolution figure, supplementary figure and video files uploaded as individual files: See our detailed guidelines for

preparing your production-ready images, <https://www.life-science-alliance.org/authors>

B. MANUSCRIPT ORGANIZATION AND FORMATTING:

Reviewer #1 (Comments to the Authors (Required)):

The manuscript presented by Tait et al. is a follow-up to a previous study published by the same group in Life Science Alliance in 2023 (Angerilli A, Tait J et al.). In that initial article, the authors demonstrated that depletion of the enzymes KMT5B and KMT5C-responsible for converting H4K20me1 to H4K20me2/3-via antisense Morpholino oligonucleotides resulted in defective ciliogenesis in multiciliated cells (MCCs). They showed that these defects were associated with downregulation of cilium- and cytoskeleton-related genes. Notably, depletion of SUV4-20H1 alone, but not SUV4-20H2, was sufficient to induce the ciliogenesis defect. This phenotype could be rescued by overexpression of the human H4K20me1 demethylase PHF8, which largely restored expression of the affected genes even in the absence of SUV4-20H enzymes and proper conversion of H4K20me1 to higher methylation states.

In the present manuscript, using the same Morpholino-based depletion approach in the *Xenopus* model, the authors unsurprisingly confirm that the observed changes in cilium/cytoskeleton gene expression are driven by loss of KMT5B. They also reconfirm that the expression of core ciliogenic regulators is unaffected, consistent with their previous double-knockdown study. Moreover, they confirm that PHF8 activity is required to rescue the ciliogenesis phenotype in the absence of SUV4-20H1, although this appears somewhat contradictory to their earlier findings, in which the phenotype could also be rescued by expression of a truncated form of *Xenopus* PHF8, potentially lacking full catalytic activity.

Finally, the authors attempt to explore the mechanism through which KMT5B regulates multiciliogenesis-related genes. Their data suggest that this effect is not due to changes in chromatin accessibility nor to a role upstream of key transcription factor cascades.

Overall, most of the conclusions presented in this new manuscript are not novel and primarily serve to confirm previous findings. The study does not offer mechanistic insights into how KMT5B influences the expression of multiciliogenesis genes. In the absence of a clear mechanistic explanation, this new article does not provide sufficient additional information to warrant publication in Life Science Alliance in its current form.

Reviewer #2 (Comments to the Authors (Required)):

The paper from Tait et al examines the mechanisms by which KMT5B downregulation affects the formation of cilia in multiciliated cells using the *Xenopus* ciliated embryonic epidermis as a model. The authors use a previously characterized morpholino to downregulate the expression of KMT5B. To address the mechanism of KMT5B effect on ciliogenesis they examined the expression of genes known to be involved in the control of MCC differentiation and the ability of multicilin to rescue the ciliogenesis defect. In addition, the authors examined the possibility that knockdown of KMT5B could lead to chromatin changes that may affect the accessibility of certain regions, which could explain the downregulation of gene expression. They also show that the previously reported rescue of the phenotype by PHF8 overexpression depends on the catalytic activity of the demethylase. The authors conclude that KMT5B may regulate multiciliogenesis through an alternative mechanism to the canonical multiciliogenic program.

Most main points are supported by the data. Specific comments and suggestions:

Figure 1A: The KMT5C morphant MCC shown in Figure 1A seems smaller than the control MCC. Is this by chance or is this a consistent phenotype? Given that a small number of genes are misregulated by knockdown of KMT5C are any of these genes known to be involved in cell size regulation or actin remodeling which is important for surface area expansion in multiciliated cells?

Figure 1B: Please clarify if the comparison is between uninjected or control morpholino injected animal caps and KMT5B/5C. Also, at what stage were the animal caps harvested for the gene expression analysis?

Figure 3: Is the lineage tracer supposed to be visible in the images?

Figure 3B: Please provide a more detailed description of the criteria used for quantifications

Figure 4: Please provide the quality control ICCs of embryos to show that the KMT5B knockdown and multicilin overexpression phenotypes are present in the batches of animal caps selected for sequencing.

Figure 4A: A defect in ciliogenesis could be caused by a defect in basal body docking or axoneme elongation. In the double knockdown reported in their earlier study (Angerilli et al. LSA 2023), morphant MCCs have defects in basal body docking and knockdown of KMT5B alone could recapitulate the phenotypic changes in multiciliated cells. However, the effect of single knockdown on basal body docking was not quantified. Could the authors examine the effect of KMT5B knockdown on basal body docking? This will further dissect the role of KMT5B in multiciliated cells and establish its role in cilia formation.

If loss of KMT5B leads to the downregulation of genes involved in ciliogenesis but not basal body amplification, could it have a similar effect in cells with a single cilium? For example, in the GRP or in the neural tube? This could show if the role of KMT5B is similar in all ciliated cells which can broaden the impact of this work. In addition, in humans, KMT5B mutations cause neurodevelopmental disorders (PMID: 35331928, 30832413, 36897941). Given that primary cilia are important for neurodevelopment could KMT5B mutations affect primary cilia formation and lead to neurodevelopmental defects?

Line 91: KMT5A instead of KMT5C

Line 152: please correct this sentence: 'ranscription factors 2 2 like foxJ1'

Line 192: concentration not volume

The quality of some of the images is not very good. Please provide higher quality images.

Reviewer #3 (Comments to the Authors (Required)):

The manuscript "Catalytic activity of KMT5B promotes ciliogenesis without affecting accessibility of ciliary genes" by Tait et al. aims to investigate the role of KMT5B in the regulation of ciliogenesis. The authors found that KMT5B promotes ciliary formation through an enzyme activity-dependent mechanism that operates in a pathway parallel to and independent of Multicilin, without altering chromatin accessibility. Knockout of KMT5B leads to significant downregulation of ciliary genes, while the expression of key ciliogenic transcription factors remains unaffected. Overexpression of the master regulator Multicilin (MCI) fails to rescue the ciliogenesis defects caused by KMT5B loss, supporting the conclusion that KMT5B functions independently of the MCI pathway. Furthermore, catalytic activity of the H4K20me1 demethylase PHF8 can revert the ciliary defects resulting from KMT5B depletion; ATAC-seq data indicate that loss of KMT5B does not significantly alter the chromatin accessibility of ciliary genes. Overall, the study presents several interesting findings, but some of the conclusions are premature and not fully supported by the presented data. Additionally, the manuscript contains multiple instances of misaligned figure citations and omitted figure labels, which undermine the reliability of the presentation. The following points represent concerns that need to be addressed:

Major points:

1. The entire study is predicated on the model that KMT5B functions via its methyltransferase activity and that rescue by PHF8 occurs through its demethylase activity. However, the authors provide no direct evidence showing changes in H4K20me1/me2/me3 levels in multiciliated cells (MCCs) or animal caps upon KMT5B knockdown or following PHF8-mediated rescue.
2. The conclusion that KMT5B functions "without affecting the accessibility of ciliary genes" relies on ATAC-seq data showing few global and significant changes in accessibility. This, however, does not fully rule out the possibility of localized, subtle, yet functionally critical alterations at specific enhancers, insulators, or other regulatory elements.
3. The observation that ciliary defects resulting from KMT5B knockdown can be rescued by catalytically active PHF8 (an H4K20me1 demethylase) suggests that the regulation of H4K20me1 is crucial in the process of ciliogenesis. The authors note stark differences in phenotypic and transcriptomic effects between KMT5B and KMT5C but only describe these without mechanistic insight. Since both enzymes catalyze the conversion of H4K20me1 to H4K20me2, what underlies their distinct functional roles?
4. While KMT5B knockdown and MCI overexpression lead to opposite changes in GO term enrichment, and although the authors show expression trends of some downstream genes under these conditions (Figure 4B) and demonstrate that MCI overexpression cannot rescue the gene expression dysregulation caused by KMT5B loss, several questions remain unresolved: What is the relationship between the gene sets regulated by MCI and those regulated by KMT5B?

Minor points:

5. Line 92 describes knockdown of KMT5A and KMT5B, while the Results section refers to KMT5B and KMT5C, which is confusing.

6. Line 104 describes that KMT5C knockdown dysregulates 83 genes, yet Figure 1B and 1C show a total of only 66 up- and down-regulated genes. This discrepancy should be clarified.
7. Line 120 cites Figure 1E in reference to tubb4a possessing a cilia-interacting C-terminal motif. However, Figure 1E appears to illustrate transcriptomic changes following KMT5C knockdown.
8. Lines 126-127 refer to GO terms such as "cilium movement" and "cilium organization" in Figure 2A, but these terms are not visibly present. It is possible that the citation should refer to Figure 2B instead.
9. The label "H1Mo" in Figures 3A and 4B is ambiguous. It should be consistently referred to as "KMT5B MO" or explicitly defined. At the same time, there is a serious layout error in the horizontal axis of Figure 3B.
10. Figure 4A is missing scale bars.
11. Line 194 directs readers to "Figure S1A" for a schematic of the method. If I understand correctly, the content of Figure S1A does not appear to match the described methodology. Instead, it seems to correspond to the RNA-seq workflow described in the subsequent paragraph.
12. There is no reference to or interpretation of Figures 4C-4E in the main text. This should be clarified by the authors.
13. Figure 5C should include browser tracks of additional key ciliary genes (e.g., foxj1, rfx2, multicilin), rather than only the three most significantly down-regulated genes.
14. In the RNA-seq analysis, the authors do not appear to clearly specify the criteria used for defining differentially expressed genes (DEGs). It is unclear whether the definition relied solely on a p-value threshold or if a combined threshold incorporating both statistical significance and a minimum \log_2 fold change was applied. Clarification on the specific thresholds used for DEG identification is essential for interpreting the results and assessing their robustness.
15. In the ATAC-seq analysis, the authors described that "We identified 118,809 total peaks, a result comparable to previous ATAC-seq experiments in *Xenopus* animal caps (Esmaeili et al., 2020)." It is unclear whether this refers to the number of peaks, their distribution, or the overall pattern of chromatin accessibility.

LSA Follow up – Rebuttal:

September 8, 2025

Re: Life Science Alliance manuscript #LSA-2025-03455

Prof. Ralph A.W. Rupp
Ludwig-Maximilians-Universität München
Biomedical Center, Molecular Biology
Grosshaderner Strasse 9
Planegg-Martinsried D-82152
Germany

Dear Dr. Rupp,

Thank you for submitting your Follow Up manuscript entitled "Catalytic activity of KMT5B promotes ciliogenesis without affecting accessibility of ciliary genes" to Life Science Alliance. The manuscript was assessed by expert reviewers, whose comments are appended to this letter. We sincerely regret the long delay faced by this manuscript due to reviewer and editor availability, and we appreciate your patience.

As you will see, reviewers diverged in their enthusiasm for this work which offers new observations related to your prior work published with LSA. We welcome submission of a suitably revised manuscript to LSA. Although we appreciate the reservations of Reviewer 1 on the advance of this work relative to the prior work, this overlap is acceptable for a Follow Up manuscript. Please attend to the distinction noted by this reviewer on rescues with human PHF8 vs truncated *Xenopus* PHF8. Reviewer 2 made several important requests to verify key observations and clarify methodology. In particular, their points related to Fig 1A (on cell size) and Fig 4A (on basal body docking) must be addressed in a revised manuscript. Reviewer 3 sought confirmation of epigenetic marks by KMT5B and PHF8, and while data for the former were shown previously we agree that confirming the effects of PHF8 rescue on histone marks should be included here. Please also discuss the remaining questions posed by this reviewer.

The typical timeframe for revisions is three months. Please note that papers are generally considered through only one revision cycle, so strong support from the referees on the

revised version is needed for acceptance.

Thank you for this interesting contribution to Life Science Alliance. We are looking forward to receiving your revised manuscript.

Sincerely,

- A letter addressing the reviewers' comments point by point.
- An editable version of the final text (.DOC or .DOCX) is needed for copyediting (no PDFs).
- High-resolution figure, supplementary figure and video files uploaded as individual files: See our detailed guidelines for preparing your production-ready images, <https://www.life-science-alliance.org/authors>
- Summary blurb (enter in submission system): A short text summarizing in a single sentence the study (max. 200 characters including spaces). This text is used in conjunction with the titles of papers, hence should be informative and complementary to the title and running title. It should describe the context and significance of the findings for a general readership; it should be written in the present tense and refer to the work in the third person. Author names should not be mentioned.
- By submitting a revision, you attest that you are aware of our payment policies found here: <https://www.life-science-alliance.org/copyright-license-fee>

B. MANUSCRIPT ORGANIZATION AND FORMATTING:

Point by point rebuttal

In blue – Reviewers' comments

In black – our response (new text underlined)

Reviewer #1:

The manuscript presented by Tait et al. is a follow-up to a previous study published by the same group in Life Science Alliance in 2023 (Angerilli A, Tait J et al.). In that initial article, the authors demonstrated that depletion of the enzymes KMT5B and KMT5C-responsible for converting H4K20me1 to H4K20me2/3-via antisense Morpholino oligonucleotides resulted in defective ciliogenesis in multiciliated cells (MCCs). They showed that these defects were associated with downregulation of cilium- and cytoskeleton-related genes. Notably, depletion of SUV4-20H1 alone, but not SUV4-20H2, was sufficient to induce the ciliogenesis defect. This phenotype could be rescued by overexpression of the human H4K20me1 demethylase PHF8, which largely restored expression of the affected genes even in the absence of SUV4-20H enzymes and proper conversion of H4K20me1 to higher methylation states.

In the present manuscript, using the same Morpholino-based depletion approach in the *Xenopus* model, the authors unsurprisingly confirm that the observed changes in cilium/cytoskeleton gene expression are driven by loss of KMT5B. They also reconfirm that the expression of core ciliogenic regulators is unaffected, consistent with their previous double-knockdown study. Moreover, they confirm that PHF8 activity is required to rescue the ciliogenesis phenotype in the absence of SUV4-20H1, although this appears somewhat contradictory to their earlier findings, in which the phenotype could also be rescued by expression of a truncated form of *Xenopus* PHF8, potentially lacking full catalytic activity.

Finally, the authors attempt to explore the mechanism through which KMT5B regulates multiciliogenesis-related genes. Their data suggest that this effect is not due to changes in chromatin accessibility nor to a role upstream of key transcription factor cascades.

Overall, most of the conclusions presented in this new manuscript are not novel and primarily serve to confirm previous findings. The study does not offer mechanistic insights into how KMT5B influences the expression of multiciliogenesis genes. In the absence of a clear mechanistic explanation, this new article does not provide sufficient additional information to warrant publication in Life Science Alliance in its current form.

We thank colleague #1 for a fair opinion and review. Publications rarely manage to address all open questions inherent to their analysis. Follow-up manuscripts, like this one here, may appear for some readers merely confirmative and unsurprising in character, while others may find the corroboration of key points helpful for future investigations.

Comment: apparent contradiction regarding PHF8 catalytic activity and cilia rescue

Our truncated *Xenopus* cDNA clone represents a splicing intermediate, which contains exons 1-8, followed by a large part of intron 8. It codes for the first 264 amino acid residues of PHF8, plus three amino acids (Val-Ser-Ser) derived from intron 8, before it terminates there at the first intronic stop codon. This short protein (aa1-267) still contains two out of three residues (H247/D249) needed for Fe(II) chelation, as well as K264, needed for 2-OG binding. It lacks the third Fe(II) chelating residue H319 (whose single significance has never been tested by mutation) and about the C-terminal half of the 2-OG binding domain. Based on this finding, we expected it to be either enzymatically compromised or completely inactive. To validate this assumption was impossible, given that even full length human PHF8 did not detectably alter the highly increased H4K20me1 levels of KMT5B-depleted embryonic chromatin (see comments by reviewer 3).

For this follow-up manuscript, we decided not to investigate how much residual activity the truncated *Xenopus* PHF8 protein might have in comparison to the full length human PHF8 protein, but to clarify, whether rescue of the ciliogenic phenotype depends on PHF8 activity or not. For this purpose, we used the two highly similar PHF8 protein constructs, kindly provided by Dr. Ramin Shiekhattar. His lab firmly established that these proteins encode active and inactive variants of PHF8 both in vitro and in vivo (Fortschegger et al., 2010). The results shown in Figure 3 now settle the issue - rescuing cilia tufts in KMT5B morphant *Xenopus* requires PHF8 catalytic activity. In hindsight, this suggests that the truncated *Xenopus* protein has some residual activity.

We have added this conclusion to the discussion on page 12, last paragraph (new text underlined):

“...only enzymatically active PHF8 restores cilia tufts in KMT5B depleted organoids. In hindsight, this result implies that the truncated *Xenopus* PHF8 protein has some residual demethylation activity, although we have not shown this. In summary...”

Reviewer #2:

The paper from Tait et al examines the mechanisms by which KMT5B downregulation affects the formation of cilia in multiciliated cells using the *Xenopus* ciliated embryonic epidermis as a model. The authors use a previously characterized morpholino to downregulate the expression of KMT5B. To address the mechanism of KMT5B effect on ciliogenesis they examined the expression of genes known to be involved in the control of MCC differentiation and the ability of multicilin to rescue the ciliogenesis defect. In addition, the authors examined the possibility that knockdown of KMT5B could lead to chromatin changes that may affect the accessibility of certain regions, which could explain the downregulation of gene expression. They also show that the previously reported rescue of the phenotype by PHF8 overexpression depends on the catalytic activity of the demethylase. The authors conclude that KMT5B may regulate multiciliogenesis through an alternative mechanism to the canonical multiciliogenic program.

Most main points are supported by the data. Specific comments and suggestions:

Figure 1A: The KMT5C morphant MCC shown in Figure 1A seems smaller than the control MCC. Is this by chance or is this a consistent phenotype? Given that a small number of genes are misregulated by knockdown of KMT5C are any of these genes known to be involved in cell size regulation or actin remodeling which is important for surface area expansion in multiciliated cells?

We did not notice a consistent difference in cell size between control morphant embryos and KMT5C morphant embryos. However, cell size can exhibit natural variation between embryos and we think this accounts for any discrepancy in cell size observed in the Figure 1A. Below are representative images from an additional two replicates, which show consistent cell size between conditions. **This additional data has been added as new Figure S1 to the supplement.**

Additionally, the few genes that were misregulated upon KMT5C knockdown do not relate to cell size or actin remodeling. This statement has been included **near the end of the first chapter on page 4:**

"...This is likely due to the overall small cohort of misregulated genes. None of these genes are involved in cell size regulation or actin remodeling. That KMT5C has such a weak effect..."

Figure S1: Knockdown of KMT5C does not lead to a change in cell size. A) 4 channel confocal images depicting single knockdowns of KMT5B or KMT5C in *Xenopus laevis* embryos. Cilia are magenta (acetylated α -tubulin), basal bodies are green (hyls1-GFP), actin meshwork and cell borders are red (phalloidin), and DNA is blue (DAPI). Embryos are injected in one of two ventroanimal blastomeres at the 8-cell stage. A') Enlarged inlays from Panel A. Scale bars = 10 μ m.

Figure 1B: Please clarify if the comparison is between uninjected or control morpholino injected animal caps and KMT5B/5C. Also, at what stage were the animal caps harvested for the gene expression analysis?

The comparison was between control morpholino injected animal caps and KMT5B/C. Embryos were harvested at NF16 (neurula stage).

Figure 3: Is the lineage tracer supposed to be visible in the images?

We can appreciate that the lineage tracer is difficult to see due to the color, intensity and similarity to the color of Alkaline Phosphatase staining of cilia tufts. The lineage tracer is visible as smooth blue staining, generally on the anterior and dorsal injected side of the embryo. For clarity we have indicated the regions of interest on the figure with yellow dash – see revised Figure 3.

Figure 3B: Please provide a more detailed description of the criteria used for quantifications

Embryos were classified by visual inspection based on prevalence of acetylated alpha tubulin staining, prevalence and size of cilia tufts, comparing the uninjected and injected embryo sides.

Figure 4: Please provide the quality control ICCs of embryos to show that the KMT5B knockdown and multicilin overexpression phenotypes are present in the batches of animal caps selected for sequencing.

Unfortunately, we do not have images of each replicate of the quality control ICCs available. Below is a representative image from one quality control replicate, as well as the scoring and quantification of the quality control from the replicates that were used for sequencing. We have incorporated the representative images and quantification into figure S3 (below), and in the text:

“embryos. (For a schematic of the MCI-induction method, including the RNA-seq workflow, see Figure S3A, for representative images and quantification of quality control embryos, see Figure S3B, C)...”

Figure S3: Overexpression of MCI leads to upregulation of multiciliogenic genes. A) Protocol for multiciliogenin overexpression. Embryos are injected in both cells of the 2-cell stage with the morpholino and mRNA of interest. They are allowed to develop until the blastula stage (NF8-9), at which point animal cap explants are dissected. At late blastula stage (NF 11), animal caps are induced with 10 μ M

dexamethasone. Animal cap explants are harvested at the neurula stage (NF16) and further processed for RNA-seq. Simultaneously, sibling embryos are injected on one cell of the 2-cell stage with constructs of interest and allowed to develop until late blastula stage (NF11). Embryos are then induced with 10 μ M dexamethasone. At tailbud stage (NF28), embryos are fixed for whole mount immunocytochemistry. B) Representative images of quality control embryos assessed to exhibit both a KMT5B knockdown phenotype and a multicilin overexpression phenotype. Embryos were stained with anti-acetylated \$\alpha\$ -tubulin for cilia tufts.

We have also included the embryo scoring for each individual replicate below:

		Up	Down	Unaffected
Rep 1:	KMT5BMo+ MCI	0	4	2
	Como + MCI	10	0	0
	KMT5B Mo	0	4	3
		Up	Down	Unaffected
Rep 2:	KMT5BMo+ MCI	0	11	0
	Como + MCI	13	0	1
	KMT5BMo	0	11	0
Rep 3:		Up	Down	Unaffected
	KMT5BMo+ MCI	0	12	0
	Como + MCI	13	0	0

Figure 4A: A defect in ciliogenesis could be caused by a defect in basal body docking or axoneme elongation. In the double knockdown reported in their earlier study (Angerilli et al. LSA 2023), morphant MCCs have defects in basal body docking and knockdown of KMT5B alone could recapitulate the phenotypic changes in multiciliated cells. However, the effect of single knockdown on basal body docking was not quantified. Could the authors examine the effect of KMT5B knockdown on basal body docking? This will further dissect the role of KMT5B in multiciliated cells and establish its role in cilia formation.

Multi-color Z-stack imaging is extremely costly and time-consuming, which had prevented us from quantifying the frequency, at which basal body clumping occurs in double knockdown embryos. We had merely stated that “the BBs in SUV4-20H1/2-depleted MCCs tend to clump”, but also noted that „most BBs, which had arrived at the cell membrane, still failed to nucleate ciliary axonemes,, (Angerilli et al., 2023). Although we view the observed delay in

BB transport to the apical membrane as part of the KMT5B/C phenotype, this second notion accurately described that morphant MCCs are also deficient in axoneme formation.

In our revised manuscript, we have added new z-stack images of several KMT5B single morphant MCCs with clumped basal bodies, staying close to the nucleus at a time, when wildtype cells are already fully ciliated (**new supplementary Figure S2**). These images demonstrate that KMT5B single knockdown can lead to a delay in BB transport, similar to what was found in double morphant cells.

We have added this confirmative finding to the first Results chapter **on page 4**:

“Knockdown of KMT5B results in fewer, shorter cilia, a depleted actin cap, and occasionally in clumped basal bodies, close to the nucleus of MCCs (Figure S2), as had been described for KMT5B/C double knockdown (Angerilli et al., 2023).”

Figure S2: Basal body clumping occurs in KMT5B single knockdown embryos. A, B, C) Representative images of three KMT5B knockdown multiciliated cells. Top images represent orthogonal views showing clumped BBs near cell nucleus; bottom row shows flattened Z-stacks. Cell in A shows that a neighbouring, uninjected MCC is fully ciliated, while the basal bodies of the KMT5B morphant cell are still located deep. Cilia are magenta (acetylated α -tubulin), basal bodies are green (hyls1-GFP), actin meshwork and cell borders are red (phalloidin), and DNA is blue (DAPI). Embryos are injected in one of two ventroanimal blastomeres at the 8-cell stage. Scale bars = 10 μ m.

If loss of KMT5B leads to the downregulation of genes involved in ciliogenesis but not basal body amplification, could it have a similar effect in cells with a single cilium? For example, in the GRP or in the neural tube? This could show if the role of KMT5B is similar in all ciliated cells which can broaden the impact of this work. In addition, in humans, KMT5B mutations cause neurodevelopmental disorders (PMID: 35331928, 30832413, 36897941). Given that primary cilia are important for neurodevelopment could KMT5B mutations affect primary cilia formation and lead to neurodevelopmental defects?

This is an interesting question, although it falls completely out of the focus of this and our previous paper. Our work has concentrated exclusively on the differentiation of cilia tufts in

MCCs and we have not made any claims that KMT5B might be important for ciliogenesis in general.

However, some unpublished data, which we have obtained in collaboration with Drs. Axel Schweickert and Thomas Thumberger from the University of Stuttgart, suggest that *Xenopus* gastrocoel roof plate (GRP) cilia are unaffected in KMT5B depleted embryos. **This new data is shown in Figure S6**, the corresponding text (underlined) starts with **the last paragraph on page 9**:

„Based on these results, one might be tempted to speculate that KMT5B acts simply downstream of MCI. For instance, it might enable FOXJ1 to form single motile cilia. For MCCs we showed previously that overexpressed FOXJ1 did not rescue cilia axonemes in KMT5B/C knockdown condition (Angerilli et al., 2023). Therefore, we have started to investigate gastrocoel roof plate (GRP) cilia (Figure S5), which are required to establish L/R-asymmetry in the embryo (reviewed by Blum et al., 2014). GRP cells express FOXJ1, but not MCI, and form single, motile cilia, whose stroke generates directional fluid flow. We injected CoMo or KMT5B-Mo into the dorso-marginal zone of the embryo and derived from this region open-face explants, which we cultured until the initial neural tube stage, when GRP cilia are fully developed. At this time point, differentiation of epidermal MCC cells is under the influence of KMT5B. We recorded scanning electron micrographs of the inner surface and analysed GRP cilia length and number (Fig. S5A-D), following an established protocol (Walentek et al., 2013). Knockdown of KMT5B had no significant effect on the formation of single, motile cilia in these cells (Figure S5E-F). These results suggest that cilia-promoting activity of KMT5B could be coupled specifically to MCI-controlled multiciliogenesis, but having no effect on FOXJ1-dependent formation of single motile cilia.“

We also include a reference to this data in the **first paragraph of the discussion on page 11**:

„Here we report that the H4K20 methyltransferase KMT5B regulates *Xenopus* multiciliogenesis in the larval epidermis, while single, motile cilia present on the gastrocoel roof plate appear unaffected.“

Figure S6: Comparing gastrocoel roof plate (GRP) cilia length and prevalence between KMT5B morphant and control morphant embryos. Scanning electron micrographs of the inner explant surface of a CoMo (A) or KMT5B-Mo (B) injected explant. Magnification 480x. Panels C and D show ImageJ-assisted assignment of cell borders and cilia position within a 320x320 μm large area from the center of the SEM images shown above. E, F) Comparative analysis of cilia length, posterior cilia length, and number of GRP cilia between KMT5B and control morphant embryos. No significant difference in any comparison ($p < 0.05$). $N = 4$ explants/condition, total cell numbers: 97/153 cells ciliated in CoMo condition; 89/118 cells ciliated in KMT5B-Mo condition.

Technical Information on GRP explants, cilia analysis and scanning electron microscopy have been added to the Materials and Methods section:

“GRP cilia and scanning electron microscopy

Gastrocoel roof plate cilia were analysed by targeted microinjection into the dorsal marginal zone of two-cell embryos, next to the newly formed cleavage plane. The prospective dorsal and ventral sides of the embryo can be often distinguished by asymmetric pigment distribution, where the dorsal side is less pigmented. Since lineage-tracing the injections by fluorescent dextrans had turned out to interfere with the SEM sample preparation, injections were restricted exclusively to embryos with unambiguous pigment asymmetry. Training rounds with fluorescent dextrans were performed until the gastrocoel roof plate was consistently targeted in all cases. Afterwards, CoMo and KMT5B-Mo were injected at the same concentration as for animal cap injections. GRP explants were microdissected at NF17 (late neural fold stage) and fixed immediately in Sorensen’s phosphate buffer with 3% glutaraldehyde for 1 hour to prevent explants from closing up. Subsequently, the explants were manually cleaned from cell debris and washed at least 3x 20min in PBS, followed by a second fixation in 1% OsO4 and stepwise dehydration into 100% ethanol. All samples were attached on a specimen stub and sputter-coated with gold. SEM images were kindly

recorded by Mrs. Beate Aschauer from the department of Prof. Dr. Ulrich Welsch (Institute of Anatomy, Medical Faculty, LMU Munich) on a Jeol:JSM-35 GF scanning electron microscope. Image orientation was determined in reference to the visible blastoporus, marking the posterior end of the explant. An area of 320x320 mm was selected from the center of the GRP image to determine cell boundaries, cilia length and position using ImageJ as published (Walentek et al., 2013)."

Line 91: KMT5A instead of KMT5C

Thank you, we have corrected this error.

Line 152: please correct this sentence: 'ranscription factors 2 2 like foxJ1'

We have corrected this typo.

Line 192: concentration not volume

Volume is actually correct in this sentence, as we keep the concentration consistent, but scale the volume depending on the size of the blastomere.

The quality of some of the images is not very good. Please provide higher quality images.

We double-checked our images and confirm that figures uploaded with our revised manuscript match LSA standard.

We thank colleague #2 for positive criticism and constructive questions, which improved the manuscript considerably!

Reviewer #3:

The manuscript "Catalytic activity of KMT5B promotes ciliogenesis without affecting accessibility of ciliary genes" by Tait et al. aims to investigate the role of KMT5B in the regulation of ciliogenesis. The authors found that KMT5B promotes ciliary formation through an enzyme activity-dependent mechanism that operates in a pathway parallel to and independent of Multicilin, without altering chromatin accessibility. Knockout of KMT5B leads to significant downregulation of ciliary genes, while the expression of key ciliogenic transcription factors remains unaffected. Overexpression of the master regulator Multicilin (MCI) fails to rescue the ciliogenesis defects caused by KMT5B loss, supporting the conclusion that KMT5B functions independently of the MCI pathway. Furthermore, catalytic activity of the H4K20me1 demethylase PHF8 can revert the ciliary defects resulting from KMT5B depletion; ATAC-seq data indicate that loss of KMT5B does not significantly alter the chromatin accessibility of ciliary genes. Overall, the study presents several interesting findings, but some of the conclusions are premature and not fully supported by the presented data. Additionally, the manuscript contains multiple instances of misaligned figure citations and omitted figure labels, which undermine the reliability of the presentation. The following points represent concerns that need to be addressed:

Major points:

1. The entire study is predicated on the model that KMT5B functions via its methyltransferase activity and that rescue by PHF8 occurs through its demethylase activity. However, the authors provide no direct evidence showing changes in H4K20me1/me2/me3 levels in multiciliated cells (MCCs) or animal caps upon KMT5B knockdown or following PHF8-mediated rescue.

This important point had already been experimentally addressed in Angerilli et al. (2023). There, Figure 1A and B show by quantitative mass spectrometry how the single or double knockdown of KMT5B and KMT5C affect the individual H4K20 methylation states in the embryo. In short, KD of either enzyme reduces H4K20me2 levels to similar extent (appr. 30%), with the double knockdown producing the strongest effect; the reduction in H4K20me2 leads to a corresponding increase in H4K20me1 levels; only KMT5C converts H4K0me2 to me3. These results have been described in Angerilli et al. (2023) on page 6, together with relevant background information.

In the discussion (page 10, Angerilli et al., 2023) we had dealt with the finding that overexpression of full length human PHF8 restores cilia tufts, but has no significant effect on H4K20me1 levels, as one might expect. However, one needs to take into account that the two enzymes interact very differently with chromatin, i.e. PHF8 acts on a quite limited portion of the genome (mostly promoters), whereas KMT5A/PR-SET7 (which methylates unmodified histone H4, newly incorporated into chromatin during last S-phase) and KMT5B operate broadly throughout the genome. We came to conclude that the local activity of PHF8 cannot be detected in the bulk of hyperaccumulated H4K20me1 in KMT5B knockdown chromatin.

This experimental data was provided in the reviewer comments section (pages 5-6), published along with the Angerilli et al. (2023) paper. We reproduce it here for convenience:

It is not clear, whether overexpression of PHF8 can be expected to change H4K20me1 abundance on the bulk chromatin level. In proliferating cells, SET8/KMT5a monomethylates during G2/M-phase most of the newly incorporated H4 proteins that are de novo synthesized in the preceding S1-phase (i.e. 50% of all H4 proteins in the nucleus). Within two to three rounds of

Full Revision

replication, about 90% of new H4 histones become monomethylated at Lysine 20 (Pesavento et al., Mol Cell Biol 2008, pmid: 17967882). Consequently, new H4K20me1 marks are expected everywhere in chromatin (for review see Jorgenson S. et al., Nucl Acids Res 2013, pmid: 23345616), in particular under SUV4-20H enzyme depletion conditions, which prevent its conversion to H4K20me2 significantly. In contrast it is known that PHF8 acts near transcriptional start sites (Qi et al., Nature 2010, pmid: 20622853; Liu et al., Nature 2010, pmid: 20622854; Asensio-Juan et al., Nucl Acids Res 2012, pmid: 22850774), and thus acts on a significantly smaller fraction of chromatin.

Despite these doubts, we have performed new experiments to measure the abundance of all four H4K20 methyl states in double morphant, PHF8 overexpressing, and PHF8-rescued embryos:

Relative abundance of H4K20 methylation states in wild-type (wt), SUV4-20H1/SUV4-20H2 double morpholino (H1H2), PHF8 overexpressing (PHF8), and PHF8-rescued double-morphant embryos (Rescue300 and Rescue900) at developmental stage NF17. Single phf8 mRNA injections were done with 900pg/embryo; in rescue experiments, phf8 mRNA was injected at 300pg/embryos and 900pg/embryo. N biological replicates are indicated in the brackets for each experimental condition; mean \pm s.e.m. “p” – propionylated (= me0, naturally unmodified). Note different y-axis scale for the me3-state in panel B.

H4K20me1 levels were not reduced by wtPHF8 (see “p”- and “me1”-columns). The marginal increase in H4K20me2/me3 marks, observed in rescued compared to double morphant samples, is not statistically significant. We also noted no difference for histone H3 methyl states on histone H3 Lysine 9 and 27 in bulk chromatin. Altogether, this points to a local effect of overexpressed PHF8, which remains masked in genome-wide elevated levels of H4K20me1.

In light of these results, we have not attempted to measure changes in H4K20 methyl levels with the new PHF8(1-459) NLS constructs. In order to inform the readers about the existing data we have now **revised our manuscript at several positions (new text underlined)**:

Abstract:

„Multiciliated cells are a specialized cell type found in the brain, reproductive and respiratory tracts of mammals and the epidermis of tadpole stage *Xenopus* embryos. KMT5B and KMT5C are histone methyltransferases that deposit the dimethyl mark on histone 4, lysine 20 (H4K20). We previously showed that KMT5B/C double knockdown downregulates H4K20me2 levels in bulk chromatin as well as transcription of cilia genes. MCCs of embryos lacking both enzymes, or only KMT5B, have depleted cilia. Here we separate the function of KMT5B...“

Last paragraph of introduction, **page 3**:

„We previously showed that double knockdown of KMT5B/C leads to a massive shift from H4K20me2 to H4K20me1 in bulk embryonic chromatin, coinciding with strong phenotypic and transcriptional effects on *Xenopus* multiciliated cells, and further, that knockdown of KMT5B alone can recapitulate the phenotypic changes in multiciliated cells (Angerilli et al., 2023).“

Discussion, **page 11 from first paragraph** down:

„...We show that KMT5B knockdown in animal caps leads to the concerted downregulation of ciliogenic genes, even when the master regulator of multiciliogenesis is overexpressed. There are several mechanisms, by which KMT5B, but not KMT5C, could promote ciliogenesis in MCCs. We discuss these below and relate them to our experimental data presented here and in Angerilli et al. (2023).

First, the enzyme may regulate cilia gene transcription by converting H4K20me1 to H4K20me2, which is clearly perturbed by KMT5B knockdown. In support of this hypothesis, we have previously shown that the ciliogenic phenotype is specific and dependent on the catalytic activity of KMT5B. We generated the phenotype using two non-overlapping morpholinos and demonstrated that only catalytically active KMT5B mRNA is able to rescue (Angerilli et al., 2023). This raises an interesting question - why does knockdown of KMT5C have no effect on cilia tufts, although it impacts H4K20me1/me2 levels to a similar extent as KMT5B (Angerilli et al., 2023)?

There is evidence that these two enzymes are not fully redundant. For example, only KMT5C catalyzes H4K20me3 (Angerilli et al., 2023); the *kmt5b* gene is essential in mice, but *kmt5c* is not (Schotta et al., 2004); amino acid sequences of the catalytic domains are higher conserved between frog and murine homologs (aa identity 98% for KMT5B, 88% for KMT5C 88%) than between the two murine genes (75%); they use different C-terminal sequences to target heterochromatin; FRAP experiments have shown that KMT5C interacts stably with heterochromatin, while KMT5B does not (Hahn et al., 2013). Indeed, the latter finding could contribute to a spatial segregation of the two enzyme activities on chromatin that may ultimately determine their different target genes and their different impact on transcriptional regulation. Until specific antibodies against KMT5B/C become available for ChIP-analysis, this hypothesis can only be approached by indirect means in the future.

That H4K20me1 is important for the ciliogenic phenotype is supported from rescue experiments with PHF8, a Jumonji C Domain (JmjC) containing histone demethylase. It is targeted to promoters by binding of H3K4me3 through its PHD finger, and the catalytic activity is carried out by the JmjC domain. PHF8 knockdown leads to a number of craniofacial and cytoskeletal defects. In PHF8 depleted Zebrafish embryos, the pharyngeal arches are reduced or absent, and brain development is impaired (Qi et al., 2010). PHF8 is also involved in regulating cell cycle progression (Liu et al., 2010). Originally, we found that overexpression of both full length human PHF8 protein and a truncated *X. tropicalis* PHF8

variant partially restored cilia tufts in KMT5B knockdown condition (Angerilli et al., 2023). Work by others has demonstrated that catalytic activity requires both Jmc-C and -D domains, corresponding to a minimal length of 447 amino acids (see Fortschegger et al., 2010; Loenarz et al., 2010). Since our Xenopus PHF8 clone encodes a much shorter protein, we considered its activity at least severely compromised. We therefore speculated that full length human PHF8 achieves ciliogenic rescue by demethylating H4K20me1, while the Xenopus variant might improve ciliogenesis by masking H4K20me1 marks (Angerilli et al., 2023). Unfortunately, we could not detect PHF8 dependent changes of H4K20me0/H4K20me1 levels by quantitative mass spec analysis to validate this assumption, most likely because the local activity of PHF8, which is recruited to promoters by its PHD-domain, remains undetectable within the much larger increase in H4K20me1 caused by the knockdown of the broadly operating KMT5B protein.

We therefore decided to readress this point by using a pair of nearly identical human PHF8 protein variants (i.e. PHF8(1-489)NLS) that are either wt active or double-mutated at H247/D249 to alanin, being enzymatically inactive (a kind gift of Dr. Ramin Shiekattar). These constructs unambiguously showed that only enzymatically active PHF8 restores cilia tufts in KMT5B depleted organoids. In hindsight, this result implies that the truncated Xenopus PHF8 protein has some residual demethylation activity, although we have not shown this. In summary, these rescue experiments support the hypothesis that excess H4K20me1 is detrimental to the ciliogenic transcription program and removal of this mark by PHF8 can partly alleviate this effect. Alternatively, PHF8 might promote a more favourable chromatin environment by demethylating other modified lysine residues on histone H3, rather than H4K20me1 (H3K9me1/me2, H3K27me1/me2, H3K36me2; see Fortschegger et al., 2010; Loenartz et al., 2010, Liu et al., 2010). However, we have not detected changes in histone H3 methylation in human PHF8 overexpressing Xenopus embryos (Angerilli et al., 2023).”

2. The conclusion that KMT5B functions "without affecting the accessibility of ciliary genes" relies on ATAC-seq data showing few global and significant changes in accessibility. This, however, does not fully rule out the possibility of localized, subtle, yet functionally critical alterations at specific enhancers, insulators, or other regulatory elements.

This comment is correct. We have altered the title accordingly:
„Catalytic activity of KMT5B promotes ciliogenesis without affecting **global chromatin accessibility**”

In addition, we acknowledge the limitations of ATAC-seq analysis in the Results section on **page 10, end of last paragraph**:

“...Overall, this does not support the hypothesis that KMT5B exerts its control on ciliogenic gene expression by affecting chromatin accessibility in a global manner. However, this analysis cannot rule out the possibility that subtle, yet functionally critical chromatin alterations at specific enhancers, insulators or other regulatory elements could cause the observed ciliogenic defect in MCCs.”

3. The observation that ciliary defects resulting from KMT5B knockdown can be rescued by catalytically active PHF8 (an H4K20me1 demethylase) suggests that the regulation of H4K20me1 is crucial in the process of ciliogenesis. The authors note stark differences in phenotypic and transcriptomic effects between KMT5B and KMT5C but only describe these without mechanistic insight. Since both enzymes catalyze the conversion of H4K20me1 to H4K20me2, what underlies their distinct functional roles?

We had already discussed this conundrum in Angerilli et al. (2023). Although both enzymes can convert mono- to di-methylated H4K20, there is considerable evidence that they are not

fully redundant. We have **revised the discussion (page 10, third paragraph)** to deal with this issue and explain the rationale behind our speculation that KMT5B and KMT5C may be spatially segregated:

“It may seem surprising that knockdown of KMT5C has no effect on cilia tufts, although it impacts H4K20me1/me2 levels to a similar extent as KMT5B (Angerilli et al., 2023). However, there is clear evidence that the two genes are not fully redundant and their functions have partly diverged. For example, only KMT5C catalyzes H4K20me3 (Angerilli et al., 2023); the kmt5b gene is essential in mice, but kmt5c is not (Schotta et al., 2008); amino acid sequences of the catalytic domains are higher conserved between frog and murine homologs (aa identity 98% for KMT5B, 88% for KMT5C 88%) than between the two murine genes (75%). Finally and potentially relevant for our study, they are known to use different C-terminal sequences to target heterochromatin; FRAP experiments have shown that KMT5C interacts stably with heterochromatin, while KMT5B interacts much more transiently (Hahn et al., 2013). KMT5C may therefore be largely trapped within heterochromatin, while KMT5B would act on both eu- and heterochromatin. Such a difference could provide a spatial segregation of the two enzyme activities on within chromatin that may ultimately determine their different target genes and their different impact on transcriptional regulation. Until specific antibodies against KMT5B/C become available for ChIP-analysis, this hypothesis cannot be directly addressed.”

4. While KMT5B knockdown and MCI overexpression lead to opposite changes in GO term enrichment, and although the authors show expression trends of some downstream genes under these conditions (Figure 4B) and demonstrate that MCI overexpression cannot rescue the gene expression dysregulation caused by KMT5B loss, several questions remain unresolved: What is the relationship between the gene sets regulated by MCI and those regulated by KMT5B?

We thank the reviewer for this profound question. We have **added new bioinformatical analyses**, which are shown in **new Figure S5** to clarify the relationship between KMT5B and MCI controlled gene sets. This data is **described on page 8, last chapter of Results**:

“Since KMT5B knockdown and MCI overexpression have opposing effects on the formation of multiciliated cells, we decided to explore whether these two conditions are regulating the same subset of multiciliogenic genes. We found that KMT5B knockdown and MCI overexpression regulate partially overlapping but largely distinct gene sets. There is a defined overlap of 154 genes between genes downregulated in the KMT5B knockdown and genes upregulated upon MCI overexpression. However, the majority of genes are regulated in only one of the two contrasts and do not show coordinated regulation across conditions (Figure S4, A).

We then performed GO analysis for biological processes on the overlapping genes between genes upregulated in MCI overexpression and downregulated in KMT5B knockdown and found that the most enriched categories were “intraciliary transport”, “cilium movement”, and “axoneme assembly”. These top categories are highly similar to the top downregulated categories in KMT5B knockdown overall, however, the most enriched upregulated GO categories for MCI-hGR overexpression overall are “centriole replication”, “centriole assembly” and “centrosome duplication” (Figure S4, B). These results argue against a simple linear model in which MCI overexpression rescues or replaces the transcriptional effects of KMT5B loss. Instead, they indicate that KMT5B and MCI act on overlapping but non-identical gene sets, with coordinated, opposite regulation confined to a limited subset of shared targets. Taken together, ...”

Technical details of GO term analysis have been included in the **revised Mat&Met section on page 18**:

“For gene ontology enrichment analysis, statistical significance was assessed using a hypergeometric framework equivalent to Fisher’s exact test (topGO, version 2.62) and ontologies provided by org.Xl.eg.db (version 3.22). RNA high-throughput sequencing data has been deposited in the NCBI GEO under the accession numbers GSE274392. “

We also include a reference to this in **the first paragraph of the discussion on page 11**:

„The phenotypic link between KMT5B knockdown and MCI overexpression is restricted, given that the two proteins act on overlapping, but non-identical gene sets.“

Figure S5: MCI-hGR overexpression and KMT5B knockdown regulate overlapping but distinct gene sets. A) UpSet plot detailing the number of overlapping and non-overlapping genes comparing KMT5B knockdown and MCI-hGR overexpression. B) Bubble plot of gene ontology (GO) analysis of genes that were downregulated upon KMT5B knockdown and upregulated upon MCI-hGR overexpression.

Minor points:

5. Line 92 describes knockdown of KMT5A and KMT5B, while the Results section refers to KMT5B and KMT5C, which is confusing.

Thank you, we have corrected this error.

6. Line 104 describes that KMT5C knockdown dysregulates 83 genes, yet Figure 1B and 1C show a total of only 66 up- and down-regulated genes. This discrepancy should be clarified. Thank you, we have clarified this point. There were 66 misregulated genes in this condition.

7. Line 120 cites Figure 1E in reference to tub4a possessing a cilia-interacting C-terminal motif. However, Figure 1E appears to illustrate transcriptomic changes following KMT5C knockdown.

Thank you for your comment. This should have referenced figure 1D, showing transcriptomic changes during KMT5B knockdown. We have changed the referenced panel and moved the reference forward for clarity:

“To our surprise, we found a tubulin protein, tubb4a, to be one of the most upregulated genes upon KMT5B knockdown (Figure 1D). However...”

8. Lines 126-127 refer to GO terms such as "cilium movement" and "cilium organization" in Figure 2A, but these terms are not visibly present. It is possible that the citation should refer to Figure 2B instead.

Thank you, we have corrected this figure reference.

9. The label "H1Mo" in Figures 3A and 4B is ambiguous. It should be consistently referred to as "KMT5B MO" or explicitly defined. At the same time, there is a serious layout error in the horizontal axis of Figure 3B.

We have updated these axis labels.

10. Figure 4A is missing scale bars.

We have added a scale bar.

11. Line 194 directs readers to "Figure S1A" for a schematic of the method. If I understand correctly, the content of Figure S1A does not appear to match the described methodology. Instead, it seems to correspond to the RNA-seq workflow described in the subsequent paragraph.

We have clarified this reference and moved it to include downstream RNA-seq workflow.

12. There is no reference to or interpretation of Figures 4C-4E in the main text. This should be clarified by the authors.

These panels have now been referenced at the appropriate locations.

13. Figure 5C should include browser tracks of additional key ciliary genes (e.g., foxj1, rfx2, multicilin), rather than only the three most significantly down-regulated genes.

Thank you for this comment. We agree that investigating changes in accessibility at key ciliogenic genes is relevant to understand whether accessibility plays a role in regulating multiciliogenesis. We have added **new browser tracks as Figure S7** and included a reference in the **revised text on page 10**: “...The most significantly changing peaks are found mostly in intergenic regions (Figure 5D) and are generally increasing in accessibility rather than decreasing. We also examined key ciliogenic transcription factors and found no difference in accessibility (Figure S6). Overall...”

Figure S7: KMT5B knockdown does not lead to decreased accessibility of key ciliogenic regulators. A) Representative ATAC-seq browser tracks of key ciliogenic transcription factors comparing KMT5B knockdown (red) against control knockdown (grey).

14. In the RNA-seq analysis, the authors do not appear to clearly specify the criteria used for defining differentially expressed genes (DEGs). It is unclear whether the definition relied solely on a p-value threshold or if a combined threshold incorporating both statistical significance and a minimum \log_2 fold change was applied. Clarification on the specific thresholds used for DEG identification is essential for interpreting the results and assessing their robustness.

Good point- Thank you! The specifics are detailed now in **revised Methods, chapter RNA-seq analysis**:

„Differential expression was assessed using DESeq2 (version 1.42.1), using the experimental batch as a random factor, and genes with an adjusted p-value < 0.1 were considered differentially expressed.“

15. In the ATAC-seq analysis, the authors described that "We identified 118,809 total peaks, a result comparable to previous ATAC-seq experiments in *Xenopus* animal caps (Esmaeili et al., 2020)." It is unclear whether this refers to the number of peaks, their distribution, or the overall pattern of chromatin accessibility.

This refers to the number of peaks. We have **updated the text** accordingly:

“...We identified 118 809 total peaks, a comparable peak number to previous ATAC-seq experiments in *Xenopus* animal caps (Esmaeili et al., 2020)...”

We thank colleague #3 for his thorough, constructive and insightful review, which has improved the manuscript considerably!

February 27, 2026

RE: Life Science Alliance Manuscript #LSA-2025-03455R

Prof. Ralph A.W. Rupp
Ludwig-Maximilians-Universität München
Biomedical Center, Molecular Biology
Grosshaderner Strasse 9
Planegg-Martinsried D-82152
Germany

Dear Dr. Rupp,

Thank you for submitting your revised manuscript entitled "Catalytic activity of KMT5B promotes ciliogenesis without affecting global chromatin accessibility". We returned this to Reviewer 2 for their evaluation. As this reviewer is satisfied with no further requests, we would be happy to publish your paper in Life Science Alliance pending final revisions necessary to meet our formatting guidelines.

MANUSCRIPT ORGANIZATION AND FORMATTING:

To avoid unnecessary delays in the acceptance and publication of your paper, please read the following information carefully. Full guidelines are available on our Instructions for Authors page, <https://www.life-science-alliance.org/authors>

- Please add a Running Title in our system.
- Please add the X and Bluesky handles of your host institute/organization, as well as your own, and/or one of the authors, in our system.
- Please be sure that the authorship listing and order are correct and match between the system and the manuscript file.
- Please include affiliations on the title page.
- Please rename "DECLARATION OF INTERESTS" to "Conflict of Interests" and "WORKS CITED" to "References."
- Please add your main and supplementary figure legends to the main manuscript text after the references section.
- Since Figure S7 has only one panel, it is unnecessary to label it as A. Please remove it from the figure and its legend.
- Please use the [10 author names et al.] format in your references (i.e., limit the author names to the first 10).
- Please include a "Data Availability" section placed after the Materials & Methods section. Please consult our guidelines at <https://www.life-science-alliance.org/manuscript-prep#format>
- Please add an Author Contributions section to your main manuscript text.
- We encourage you to revise the figure legend for Figure S3 such that the figure panels are introduced in alphabetical order.
- Please include details on the microscope (lasers/illumination and objectives, with magnification and NA) in the methods for immunocytochemistry.

We welcome submissions of potential cover images for the issue of LSA in which your work would appear. If you have high quality images associated with this work, please feel free to email these, with a caption, to the journal office.

LSA encourages authors to provide a 30-60 second video where the study is briefly explained. We will use these videos on social media to promote the published paper and the presenting author (for examples, see <https://docs.google.com/document/d/1-UWCfbE4pGcDdcgzcmiuJI2XMBJnxKYeqRvLLrLSo8s/edit?usp=sharing>). Corresponding or first-authors are welcome to submit the video. Please submit only one video per manuscript. The video can be emailed to contact@life-science-alliance.org

FINAL FILES:

The following items are required for acceptance.

The license to publish form must be signed before your manuscript can be sent to production. A link to the license to publish form will be available to the corresponding author only. Please take a moment to check your funder requirements.

Thank you for your attention to these final processing requirements. Please revise and format the manuscript and upload materials as soon as you are able.

Thank you for this interesting contribution to the literature. We look forward to publishing your paper in Life Science Alliance.

Sincerely,

Reviewer #2 (Comments to the Authors (Required)):

In this revised manuscript the authors have sufficiently addressed the reviewer comments.

March 19, 2026

RE: Life Science Alliance Manuscript #LSA-2025-03455RR

Prof. Ralph A.W. Rupp
Ludwig-Maximilians-Universität München
Biomedical Center, Molecular Biology
Grosshaderner Strasse 9
Planegg-Martinsried D-82152
Germany

Dear Dr. Rupp,

Thank you for submitting your Follow manuscript entitled "Catalytic activity of KMT5B promotes ciliogenesis without affecting global chromatin accessibility". It is a pleasure to let you know that your manuscript is now accepted for publication in Life Science Alliance. Congratulations to you and your coauthors on this work which I found very interesting.

Your article will publish open access upon publication under a CC-BY license.

DISTRIBUTION OF MATERIALS:

Again, congratulations on a very nice paper. From our prior correspondence I am glad that the review process was constructive and that you are satisfied overall with our handling of your paper. We look forward to future exciting submissions from your lab.

Sincerely,
